# Multiciliated cells use filopodia to probe tissue mechanics during epithelial integration in vivo

Guilherme Ventura[1,4], Aboutaleb Amiri [2,4], Raghavan Thiagarajan [1], Mari Tolonen[1], Amin Doostmohammadi [3] ✉ & Jakub Sedzinski [1] ✉

During embryonic development, regeneration, and homeostasis, cells have to migrate and physically integrate into the target tissues where they ultimately execute their function. While much is known about the biochemical pathways driving cell migration in vivo, we are only beginning to understand the mechanical interplay between migrating cells and their surrounding tissue. Here, we reveal that multiciliated cell precursors in the *Xenopus* embryo use filopodia to pull at the vertices of the overlying epithelial sheet. This pulling is effectively used to sense vertex stiffness and identify the preferred positions for cell integration into the tissue. Notably, we find that pulling forces equip multiciliated cells with the ability to remodel the epithelial junctions of the neighboring cells, enabling them to generate a permissive environment that facilitates integration. Our findings reveal the intricate physical crosstalk at the cell-tissue interface and uncover previously unknown functions for mechanical forces in orchestrating cell integration.

As cells migrate, push or pull on their neighbors in a tissue, they are embedded in a complex 3D environment that continuously exposes them to diverse mechanical stimuli[1]. The combination of biophysical and theoretical methods together with recent advances in measuring mechanical stresses in vivo has revealed how cells mechanically interact with their passive environment, for example, by sensing the stiffness of the extracellular matrix (ECM)[2–4]. These strategies have also been used to describe how mechanical inputs drive cellular behaviors in the plane of the epithelial monolayers[5], such as the forces driving apical constriction[6], convergent extension[7], or epithelial cell extrusion[8]. Despite these advances, we know comparatively little about the mechanical crosstalk at the interface of migrating cells and their surrounding tissues, which underlies a range of developmental, regenerative, and pathological events, e.g., during epithelialization, homeostatic cell renewal, and cancer cell invasion[9–12], respectively.

Common to many of these cell-tissue interactions is the movement and subsequent integration of new cells within the overlying epithelium (Fig. 1a). In many multi-layered tissues, new cells originate from basally-positioned progenitors that move apically and join the existing epithelial sheet[10,11,13]. Similarly, during the formation of the mucociliary epithelium in the amphibian *Xenopus* embryo, successive waves of precursor cells move from the basal into the superficial epithelial layer[14–17]. The first wave of migrating cells is composed of multiciliated cell (MCC) precursors, which integrate into the superficial epithelial layer composed of mucus-producing goblet cells[14,18,19] (Fig. 1a and Supplementary Fig. 1a). This multistep process, collectively known as radial intercalation, requires a complex interplay between the migrating cell and the neighboring tissue and serves as a model to study the broader process of cell integration in vivo[20–23] (Supplementary Fig. 1a). Previous studies have shown that MCCs move into the superficial layer and integrate within the tissue by pushing the neighbors aside as they expand their apical domains[18,24] (Fig. 1a and Supplementary Fig. 1a). Prior to integration, however, MCCs insert at the epithelial vertices formed by three goblet cells, commonly referred to

[1]The Novo Nordisk Foundation Center for Stem Cell Medicine (reNEW), University of Copenhagen, Blegdamsvej 3B, 2200 Copenhagen, Denmark. [2]Max Planck Institute for the Physics of Complex Systems, Nöthnitzer Str. 38, 01187 Dresden, Germany. [3]The Niels Bohr Institute, University of Copenhagen, Blegdamsvej 17, 2100 Copenhagen, Denmark. [4]These authors contributed equally: Guilherme Ventura, Aboutaleb Amiri. ✉e-mail: doostmohammadi@nbi.ku.dk; jakub.sedzinski@sund.ku.dk

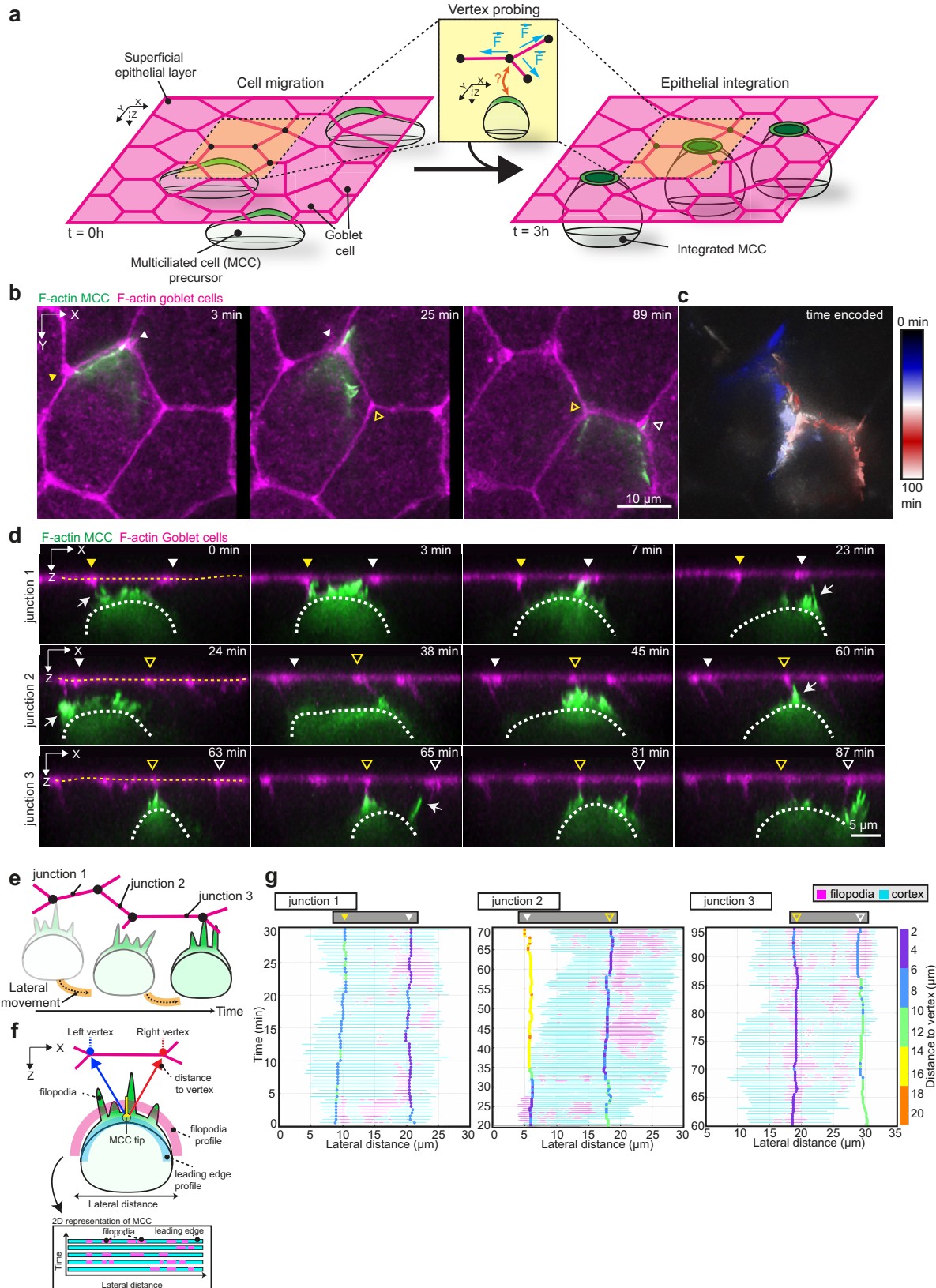

as tricellular junctions[14] (Supplementary Fig. 1a). These structures act as the susceptible positions within the epithelium that facilitate the most efficient integration within the tissue[14].

Epithelial vertices have recently been identified as key structural components integrating both biochemical and mechanical cues within epithelial sheets, and as being responsible for directing cell division and cell migration[25–27]. Of particular interest, epithelial vertices have been described as mechanical hotspots within the tissue as they sustain the tension generated by the connecting epithelial junctions[28–30] (Fig. 1a, inset). However, how this distinct mechanical feature of epithelial vertices contributes to cell integration and how radially intercalating cells select which vertex to insert into is unknown.

**Fig. 1 | Multiciliated cells probe the neighboring environment during integration. a** Schematics representing multiciliated cell (MCC) integration into the superficial epithelium. MCCs (in green) migrate into the superficial epithelium ($t = 0$ h) to integrate at the epithelial vertices formed by the neighboring goblet cells (in magenta) ($t = 3$ h). Inset depicts vertex probing by a single MCC. Epithelial vertices (black dots) form hotspots of mechanical tension as connecting junctions (magenta) pull on the vertex (blue arrows). **b–g** Dynamics of vertex probing by MCCs. MCC expresses α-tubulin::LifeAct-GFP (pseudo-colored in green) while goblet cells express nectin::utrophin-RFP (pseudo-colored in magenta). Yellow and white arrowheads, with and without fill, mark the position of different vertices and white arrows point to filopodia. **b** Image sequence from *XY* projection of MCC moving in between the overlying goblet cells. Scale bar: 10 μm. **c** Temporal-color-coded *XY* projection of MCC in **b**. **d** Orthogonal (*XZ*) projections of MCC in **b** used for filopodia dynamics analysis. White dotted lines outline the MCC contour and yellow dotted lines outline the top of the superficial epithelium. Arrowheads mark the position of the epithelial vertices. Scale bar: 5 μm. **e** Schematics representing the lateral movement of integrating MCCs. **f** Schematic representation of the main components of the filopodia analysis pipeline (see "Methods"). **g** Filopodia analysis of integrating MCC from **d**. The relative position of F-actin protrusions (filopodia, magenta) extended by a single MCC from its leading-edge (cyan, each line representing an individual time point) and the overlying epithelial vertices (vertical tracks, color-coded for the distance to vertex (between the MCC tip and the left or right vertex)) during MCC lateral movement. Arrowheads mark the position of the epithelial vertices as in **d**.

In this study, we reveal how MCC progenitors interact with the epithelial vertices to integrate within the overlying superficial epithelium of the amphibian *Xenopus* embryo. Using quantitative in vivo imaging, we find that MCCs use F-actin-based protrusions to pull at the overlying vertices of the neighboring goblet cells. By developing a minimal theoretical model of cell integration, we show that the integrating cells use pulling to read the vertex stiffness, a measure of vertex resistance to displacement upon pulling. We find, both in silico and in vivo, that MCCs prefer to insert at stiffer vertices, which correspond to the vertices made by four- and higher number of epithelial junctions. These higher-fold vertex configurations facilitate vertex opening and thus cell integration within the epithelium. We further show that such higher-fold configurations are created by the MCCs as they remodel the neighboring tissue. At a molecular level, MCC integration depends on the activity of the vertex protein LSR (lipolysis-stimulated lipoprotein receptor) and the molecular motor non-muscle myosin II. Defects in either LSR or myosin II function within MCCs lead to integration failures. Our results provide a mechanistic understanding of how migrating cells use the epithelial vertices to perceive, remodel and integrate within the surrounding epithelium.

## Results

### Multiciliated cells form dynamic filopodia targeting the epithelial vertices

To explore the potential mechanical crosstalk between the epithelial vertices and the integrating cells, we first characterized the dynamics of migrating MCCs as they begin to move into the superficial epithelial layer[14]. Using cell-type-specific α-*tubulin* and *nectin* promoters[18], we expressed the actin biosensors LifeAct-GFP and Utrophin-RFP in the MCCs and the neighboring goblet cells, respectively. Three-dimensional (3D) time-lapse imaging revealed that MCCs accumulated filamentous actin (F-actin) at their leading edge, from which they extended finger-like protrusions as they ascended apically (Fig. 1b, c). These F-actin-rich filopodia were dynamic and pointed at the cell junctions overlying the MCCs. We observed that, during this behavior, MCCs interacted with multiple vertices as they moved laterally unrestricted by other neighboring MCCs (Fig. 1d, Supplementary Fig. 1b, and Supplementary Movie 1). To understand whether filopodia are randomly assembled along the leading edge or if they are directed to specific positions in the tissue, such as the epithelial vertices, we established an image analysis pipeline to quantify F-actin protrusion activity and position within the MCC's leading-edge during integration (Fig. 1e, f and Supplementary Fig. 1c). Our analysis revealed that while cells extended filopodia along their entire leading edge, filopodia were consistently enriched at vertices (Fig. 1g, Supplementary Fig. 1d and Supplementary Movie 2). Remarkably, we observed that cells did not interact with one single vertex at the time, but extended filopodia at several vertices in their vicinity and often moved closer to a neighboring vertex after its initial probing (Fig. 1g, Supplementary Fig. 1d, and Supplementary Movie 2). Combined, these results show that filopodia are consistently formed at the positions of the leading edge closest to the vertices, strongly suggesting that filopodia guide cell

movement by transmitting spatial information from the surrounding goblet cells. Interestingly, filopodia are known to exert pulling forces in the ECM to probe its mechanical properties[31,32]. Therefore, we asked whether MCC-generated filopodia have a similar role in sensing the mechanical features of the overlying epithelial vertices.

### LSR mediates filopodia formation and integration of multiciliated cells

To further examine the relationship between the filopodia generated by the MCCs and the epithelial vertices, we imaged one of the main structural components of the epithelial vertices: the lipolysis stimulated lipoprotein receptor (LSR/angulin-1)[33]. LSR extends basolaterally to form a string-like structure[34] (Supplementary Fig. 2a). We visualized LSR-3xGFP[35] in the goblet cells and performed dynamic imaging of integrating MCCs expressing LifeAct-RFP (Fig. 2a–c and Supplementary Movie 3). We confirmed that the vertices were consistently targeted by dynamic filopodia, which formed temporary contacts with the LSR strings (Fig. 2a–c, Supplementary Fig. 2a, b and Supplementary Movie 3). Upon multiple cycles of contact formation and retraction between filopodia and the LSR string, filopodia concentrated around the LSR structure, leading to the accumulation of F-actin within the MCC cortex at the selected vertex (Supplementary Fig. 2f). These results reinforce the notion that close contact between the MCC and the vertices, mediated by the filopodia, precedes cell insertion.

We surprisingly found that LSR also localized at the tips of filopodia, where it was maintained during filopodia extension and retraction (Fig. 2d, Supplementary Fig. 2d, g, and Supplementary Movie 4). We then hypothesized that LSR is recruited to the leading edge of the integrating MCCs where it could directly interact with the LSR localized at the epithelial vertices. Using the cell-type-specific expression of LSR, we observed colocalization of the different LSR expressed by the two cell types (Supplementary Fig. 2e). Thus, our data demonstrate that MCCs use filopodia to directly interact with vertices through potential LSR-LSR-mediated contacts.

We observed that LSR knockdown using two different morpholinos in MCCs dramatically impaired epithelial integration by blocking the apical emergence of MCCs (86% of all LSR MO#1 and 73% of all LSR MO#2 cells) (Supplementary Fig. 3a, b, g, h and Supplementary Movie 5). LSR-depleted MCCs were able to reach the superficial layer (45% of all LSR MO cells), while others disappeared back into the basal layer, suggesting that a strong attachment to the overlying vertices might be required to stabilize the MCCs in the superficial layer (27% of all LSR MO cells). Notably, 14% of LSR MO cells died after failing to integrate the superficial layer (Supplementary Fig. 3b, d, h). LSR-depleted cells also failed to sustain any prominent filopodia growth, in contrast with the control cells (Supplementary Fig. 4a–f). This is in line with a previously described role for LSR in actin cytoskeleton regulation in epithelial cells (Supplementary Fig. 3e, f)[36,37]. Conversely, LSR overexpression in MCCs induced the ectopic formation of filopodia from expanding apical domains (Supplementary Fig. 2h and Supplementary Movie 6). Altogether, these findings show that LSR regulates the actin cortex dynamics and filopodia activity in the MCCs, which

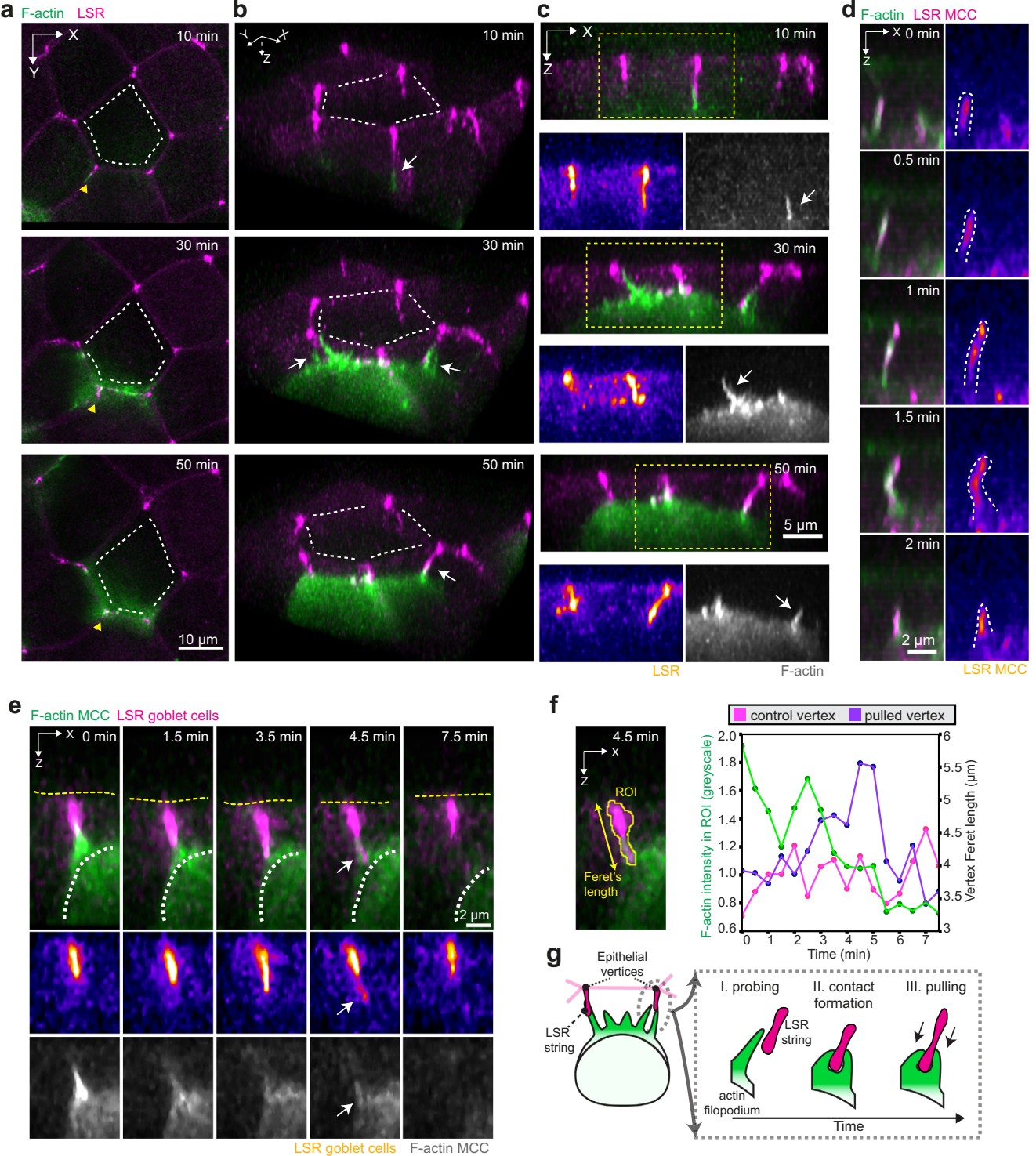

**Fig. 2 | Integrating MCCs pull on the epithelial vertices. a–d** Filopodia interact with epithelial vertices as MCC moves into the superficial epithelium. The actin cortex of MCCs is labeled with α-tubulin::LifeAct-RFP (pseudo-colored in green in composite images and gray as a separate channel). **a** Image sequence of integrating MCC interacting with overlying vertices. Epithelial vertices are labeled with lipolysis stimulated lipoprotein receptor tagged with x3GFP (LSR, pseudo-colored in magenta in composite images and fire as a separate channel). Scale bar: 10 μm. White dotted lines outline overlying junctions. The yellow arrowheads depict the orientation used for 3D rendering in **b. b** 3D rendering of **a**, with MCC forming contacts with different vertices (marked by white arrows). White lines outline overlying junctions. **c** Orthogonal (*XZ*) projections of **a**, depicting the attachment between filopodia (marked by white arrows) and vertices. Yellow boxes mark insets

for separate channels. Scale bar: 5 μm. **d** Close-up of LSR-GFP (pseudo-colored in magenta in composite and fire as a separate channel) localization within a growing and retracting filopodium, visualized by F-actin marker (LifeAct, pseudo-colored in green). LSR is visualized by expressing α-tubulin::LSR-GFP. Scale bar: 2 μm. **e** Orthogonal (XZ) projections of filopodium pulling on the epithelial vertex (marked by white arrows). The epithelial vertex is marked by expressing nectin::LSR-GFP. The white dotted line outlines the MCC contour and the yellow dotted line outlines the apical surface of the superficial epithelium. Scale bar: 2 μm. **f** Quantification of vertex pulling from **e**. The MCC F-actin intensity (green) and vertex length during one event of vertex pulling (purple) and for a non-pulled vertex (magenta). **g** Schematics representing MCC probing and vertex pulling.

are, in turn, required for successful cell integration within the epithelium (Supplementary Fig. 2i).

In addition to dissecting how the integrating MCCs interact with the epithelial vertices, we observed that as filopodia contact with the LSR string, they are able to pull on the epithelial vertices (Fig. 2e, f, Supplementary Fig. 2a, b and Supplementary Movie 7 and 8). When pulled by filopodia, the vertex underwent a quick elongation followed by retraction as the filopodia detached (Fig. 2f and Supplementary Fig. 2c). We hypothesized that MCCs could exploit the ability to pull on the vertices of the overlying epithelium to probe for points in the tissue, which can be used for their integration (Fig. 2g).

## MCCs probe the local vertex stiffness of the overlying epithelium

To provide a quantitative understanding of how MCCs could probe the mechanical landscape of their overlying epithelium, we developed a minimal theoretical model of the integrating cell–epithelial tissue interaction based on a vertex-based model[38] (see Supplementary Note 1). To simulate filopodia-induced pulling in tissue of heterogeneous line tension, we sequentially applied an out-of-plane force of a fixed magnitude, $f$, at each vertex while maintaining all other vertices in the plane and measured the out-of-plane displacement, $\delta$, of the vertex as a result of the applied force (Fig. 3a). By repeating this step for all the vertices in the epithelial sheet, we obtained the map of local *vertex stiffness* $K_\delta = f/\delta$ (Fig. 3a, b). We found that the vertex stiffness $K_\delta$ against out-of-plane pulling force $f$ correlates positively with the number and line tension of connecting junctions that constitute the vertex (Fig. 3b, d, see Supplementary Note 1). Assuming that each junction has a finite line tension, simple scaling arguments suggest that $K_\delta \sim z\langle\gamma\rangle$, where z denotes the number of junctions at the vertex and $\langle\gamma\rangle$ is the average line tension of those junctions. We further explored the susceptibility of different vertices to integration by inserting a cell of an initial area $A_{in}$, much smaller than the preferred cell area $A_0$, at

each vertex, one at a time. In this model, the pressure difference between the inserting cell and its neighbors expands the cell toward the target area, which is resisted by the inserting cell's cortical tension and enhanced by line tensions of connecting junctions constituting the vertex (Supplementary Fig. 12)[18]. The model predicted a high propensity of successful integrations at the vertices where local stiffness is greatest (Fig. 3c–e). This is due to the higher number of adjacent cell junctions (constituting a vertex) and, consequently, a more significant sum of line tensions pulling at the vertex, enhancing the expansion of the MCC's apical domain within the epithelial sheet (Fig. 3b, c). Moreover, the existence of heterogeneous line tension induces the formation of higher fold vertices, where four or more cells meet[39], and the model predicted that fourfold and higher-fold vertices should open up easier than the predominant threefold vertices in the overlying tissue (Fig. 3f) (see Supplementary Note 1). Furthermore, the model predicted that at the onset of the integration of a cell with an initial area $A_{in}$ at a $z$-fold vertex, each connected junction pulls on the cell with an opening force $f_o$ leading to an expansion pressure $P_o = zf_o/A_{in}$ (see Supplementary Note 1). Consequently, higher-fold vertices ($z > 3$) are more likely to open up than the predominant threefold vertices in the overlying vertex model tissue (Fig. 3f, see inset).

To verify the model predictions experimentally, we quantified the evolution of vertex fold number, vertex stiffness, and the propensity of MCCs to integrate at a particular vertex type in vivo. Informed by the in silico model prediction that vertex stiffness directly depends on the tension of junctions connecting the vertex (Fig. 3b, d), we measured the sum of line tensions of the junctions connected to vertices as a proxy of vertex stiffness in vivo. Specifically, as a readout of junctional tension, we quantified myosin II intensity using a non-muscle myosin II A-specific intrabody (SF9-3xGFP, for simplicity referred as myosin II), which has been previously used as a proxy for active myosin II[40] (Fig. 4a, b and Supplementary Movie 9). We validated this approach in

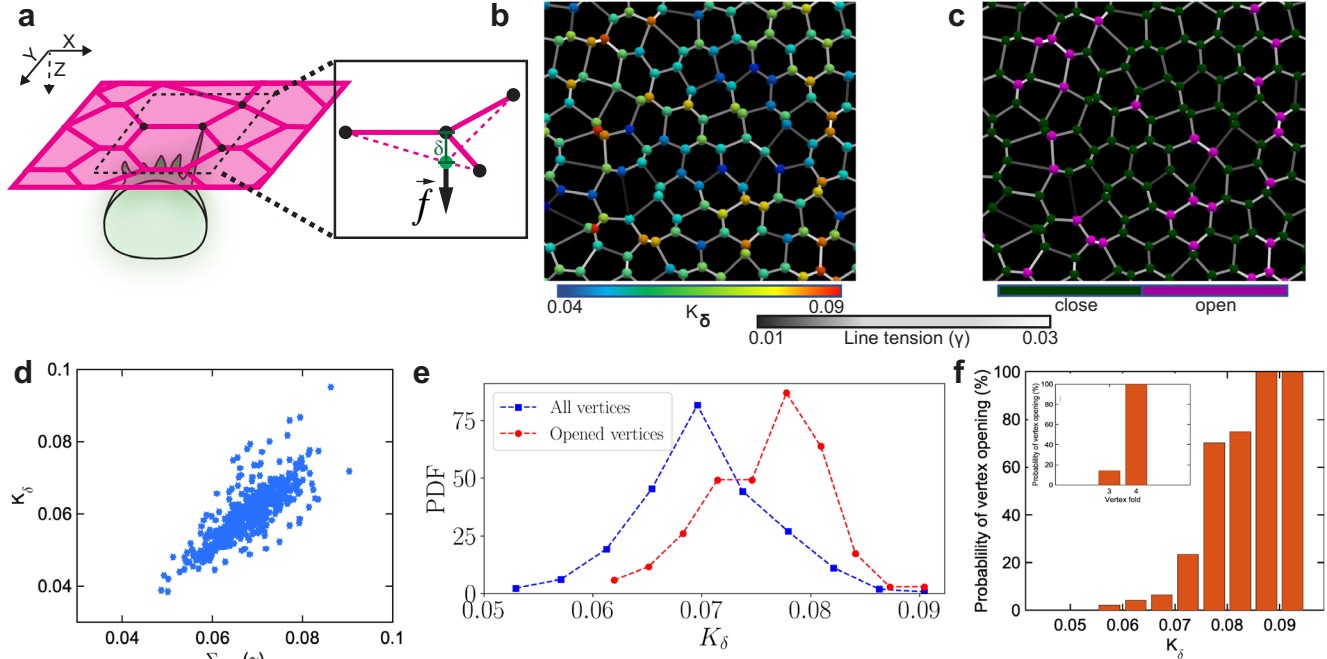

**Fig. 3 | Vertex model predicts that vertices with higher-fold and net line tension provide preferred spots for MCC integration. a** Schematics represent the out-of-plane force ($f$) exerted by an integrating MCC on an epithelial vertex inducing an out-of-plane displacement ($\delta$). **b** Representative snapshot of the simulated cellular network. The colormap on junctions indicates line tension ($\gamma$), while the vertices are color-coded according to their stiffness ($K_\delta$). **c** Representative snapshot of the cellular network illustrating propensity of vertices to open upon cell integration.

Purple (dark green) vertices mark vertices with successful (failed) insertions.
**d** Vertex stiffness increases with the increasing sum of line tensions at each vertex. **e** The probability density function (PDF) of the vertex stiffness distributions for all vertices in the simulated cellular network (blue) and the vertices with successful integration events were (red). **f** The probability of vertex opening for varying vertex stiffness values. The inset shows the opening probability for threefold vs. fourfold vertices.

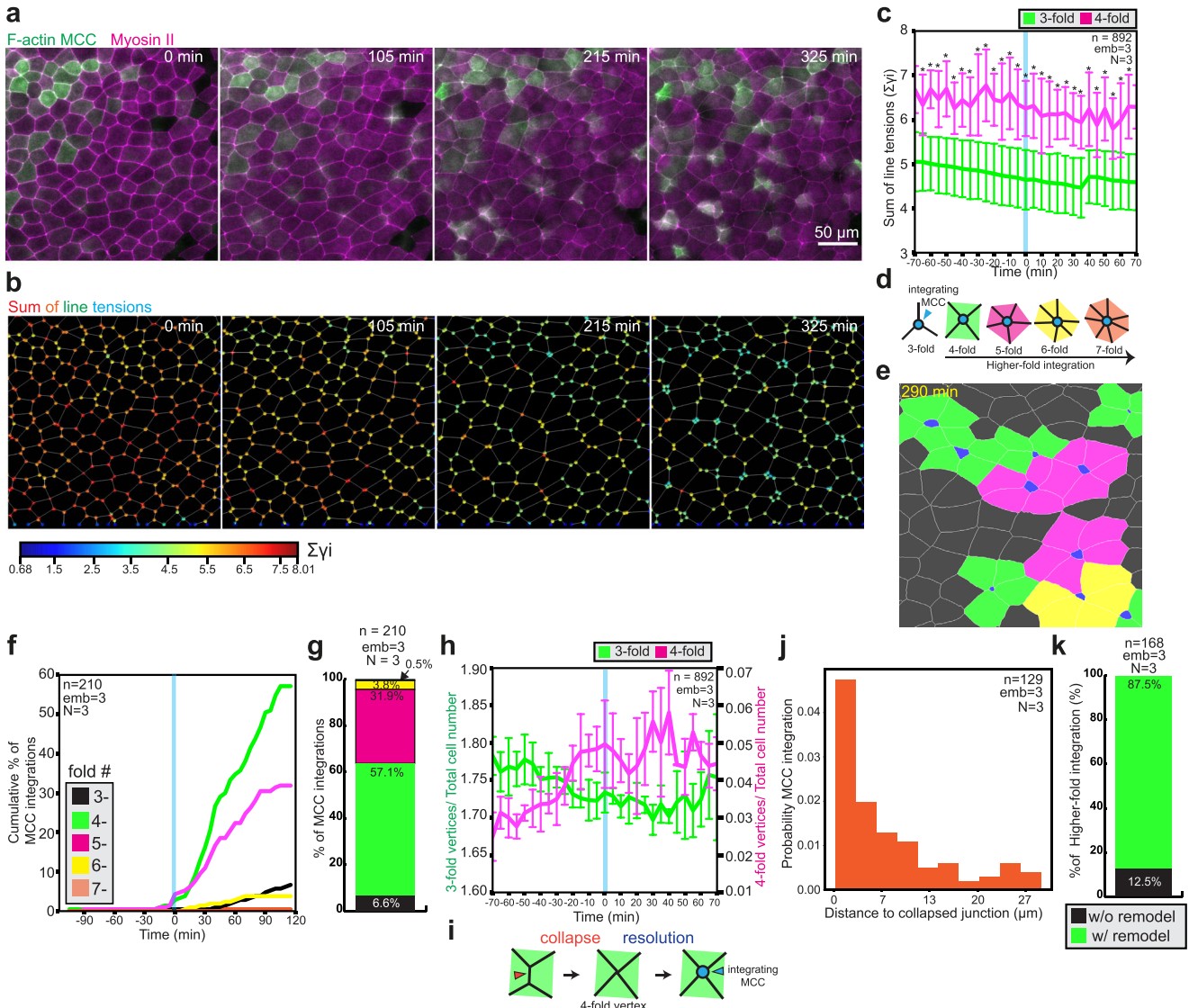

**Fig. 4 | MCCs integrate at higher-fold, stiffer vertices. a** Snapshots of superficial epithelium throughout MCC integration. MCCs and myosin II are labeled by expressing α-tubulin::LifeAct-RFP (pseudo-colored in green) and the myosin intrabody SF9-3xGFP (pseudo-colored in magenta), respectively. Scale bar: 50 μm. **b** Sum of line tensions across time color-coded from low (blue) to high tensions (red) extracted from **a**. **c** Sum of line tensions for threefold vertices (green) and fourfold vertices (magenta) during MCC integration (from $n = 892$ cells from 3 embryos, $N = 3$ experiments). Data show mean ± SD, asterisks represent significant statistical differences between datasets. Two-tailed Mann−Whitney test *$p = $[2.00E −13, 0.00176]. **d**−**g** Quantification of MCC integration according to the number of neighboring goblet cells. **d** Schematics representing higher-fold integrations, color-coded according to the number of neighboring cells. **e** Segmented image depicting higher-fold cell integrations, color-coded according to **d**. Apically expanding MCCs are marked in blue. **f** Cumulative percentage of MCC integrations across time. $T = 0$ marks the onset of MCC integration (defined as the 1% addition of new MCCs into the epithelium) ($n = 210$ cells from 3 embryos, $N = 3$ experiments). **g** Total percentage of MCC integrations according to the number of neighboring cells. ($n = 210$ cells from 3 embryos, $N = 3$ experiments). **h** Time evolution of threefold vertices (green) and fourfold vertices (magenta) number (from $n = 892$ cells from 3 embryos, $N = 3$ experiments). $T = 0$ (blue line) marks the onset of MCC integration (defined as the 1% addition of new MCCs into the epithelium). Data show mean ± SEM. **i** Schematics representing junction collapse (orange arrowhead) into a fourfold vertex and resolution by MCC integration (blue arrowhead). **j** The probability of MCC integration as a function of distance to the location of the closest junction collapse. **k** Relative percentages of MCC integration with (green) and without (black) junction remodeling ($n = 168$ cells from 3 embryos, $N = 3$ experiments).

---

our model system by performing laser ablation of epithelial junctions followed by measuring their recoil velocities, and quantifying active phosphorylated myosin II (pmyosin II) at the epithelial junctions (Supplementary Fig. 5a−e, Supplementary Movie 10, and "Methods")[41]. These experiments showed that shorter junctions have higher recoil velocities, pmyosin II and myosin II intensities. (Supplementary Fig. 5d, e). Consistent with our model's predictions, we found that vertex stiffness scaled up with the vertex fold number and that it remained relatively constant throughout cell integration (Fig. 4c). Additionally, by performing laser ablation of junctions constituting threefold and fourfold vertices, we showed that individual junction tensions are similar between lower and higher-fold vertices, further supporting our

model's predictions that vertex stiffness scales up with the number of junctions forming a vertex (Supplementary Fig. 5f). Next, we manually scored the timing of MCC insertion and quantified the number of neighboring goblet cells to see if MCCs prefer higher-fold integrations (Fig. 4d, e and Supplementary Movie 9). We found that, cumulatively, integrations at fivefold, sixfold, and sevenfold vertices accounted for 36.2% of all integration events; fourfold integrations accounted for 57.1% of all events, whereas threefold integrations represented only 6.6% of all integrations, consistent with previous work[23] (Fig. 4f, g). Altogether, our experimental results validate the prediction from the model that MCCs preferentially integrate at higher-fold vertices that accommodate overall higher junctional tension and consequently are

stiffer than threefold vertices (Fig. 4g). Nevertheless, it remains unclear how such integration points into the tissue are being formed.

To address this question, we tracked how the number of threefold and higher-fold vertices evolves throughout the process of cell integration. Interestingly, the quantitative analysis of the movies revealed a decrease in the number of threefold vertices and the concomitant formation of fourfold vertices, which preceded the onset of MCCs insertion (Fig. 4h). This suggests an active mechanism where a junction collapses to bring two threefold vertices together, forming one fourfold vertex, resembling T1 transition in the *Drosophila* embryo[42]. Such fourfold vertices are then the preferred insertion point for MCCs, which are resolved as MCCs expand their apical domains (Fig. 4i). Surprisingly, when we quantified the distance between the position of junction collapse and the MCC integration events, we observed that MCCs predominantly integrated in the immediate vicinity of collapsing junctions (Fig. 4j and Supplementary Fig. 5g, h). The proximity between the junction collapse and MCC integration events suggested that these two events could be coupled. Analysis of the time-lapse movies supported this hypothesis as 87% ($n = 147$ of 168 events) of MCC integration events in the high-fold vertices coincided with junction collapse (Fig. 4k). Moreover, we found that junction collapse is initiated when MCCs reach the superficial layer (Supplementary Fig. 5i–k). Combined, our data suggest that the integrating MCC could be involved in the process of generating the ideal positions for their integration.

## MCCs actively remodel the neighboring epithelia to induce higher-fold vertices

To address the possibility that junction collapse is dependent on the integrating MCC, we explored the mechanisms underlying the formation of high-fold vertices. Given that junction collapse has been extensively described to be driven by cell junction contraction and reliant on non-muscle myosin II[42–44], we first tested whether the formation of higher-fold vertices is driven by myosin II in the goblet cells (Supplementary Fig. 6a). Surprisingly, we observed no evident accumulation of myosin II prior to junction collapse (Fig. 5a, b; Supplementary Fig. 6b, and Supplementary Movie 11). Instead, we found that myosin II was only accumulated after the junction started collapsing, suggesting that myosin II accumulated in response to an external stimulus that promotes the initiation of junction collapse. To study this intriguing possibility, we next analyzed the relative position of the integrating cell at the onset of junction collapse. Interestingly, junction collapse followed the formation of stable contacts between the MCC and the vertices (Fig. 5a, e, f, and Supplementary Fig. 6g, i). Moreover, using F-actin intensity in the MCC as a proxy for the proximity of the MCC to the overlying junction, we observed that the increase in the MCC F-actin intensity preceded both the start of junction collapse (Fig. 5c, g and Supplementary Fig. 6c, h, j) and the accumulation of junctional myosin II in the goblet cells (Fig. 5d and Supplementary Fig. 6d). This, together with our data on the onset of junction collapse, suggests that MCCs initiate the active remodeling of the superficial epithelial layer.

However, how does an MCC trigger junction collapse? To address this question, we envisioned two alternative mechanisms for how MCCs trigger junction collapse: (i) indirect, where MCCs induce the goblet cells to collapse the junctions after the two cell types establish close contact, and (ii) direct, where continuous pulling at the vertices by the MCC triggers the neighbor junction remodeling. To distinguish between these two alternatives, we reasoned that if the vertices are actively being pulled by the integrating MCC, then we should observe quick vertex retraction whenever the integrating cell loses contact. To test this hypothesis, we tracked junction collapse in the early stages of integration, when cells are able to freely interact with multiple vertices. We observed that junction remodeling could be quickly reverted whenever the integrating MCC lost direct contact with one of the overlying vertices (Supplementary Fig. 6e–k, Supplementary Movie 12, and Supplementary Movie 13).

To better understand how this pulling mechanism by the integrating cells could trigger junction collapse, we returned to our theoretical model and conducted a numerical experiment in which we probed the response of junctions—one at a time—to external contraction (Fig. 5h). In order to quantify the junction fate under tensional perturbation, we define $l_f/l_i$ as the order parameter characterizing the ratio of the particular junction's length after applying a positive perturbation to its initial junctional tension (Fig. 5h). The simulation results showed that fourfold vertices were formed by the collapse of junctions with sufficiently short length and sufficiently large tension (Fig. 5h). Experimental data showed a similar trend for the collapse of junctions with varying junction length and tension (Fig. 5i). Moreover, compared to the line tension, the initial length of the junction plays a more dominant role in determining whether a junction collapses or not, as the probability of junction collapse is more sensitive to changes in the initial junction length (Fig. 5h–j). Combined, these results show that as integrating MCCs enhance the effective tension of a junction in the superficial layer by pulling on its vertices, they can induce junctional re-arrangements to form higher-fold vertices, which in turn are the preferential sites for the MCCs to integrate. Recent work has, however, described how higher-fold vertices are actually juxtaposed threefold vertices[45]. To address this possibility, we performed live super-resolution imaging of rosette formation events. Our data suggests that during higher-fold vertex formation several vertices collapse into a hybrid structure composed of the integrating MCC and the surrounding goblet cells (Supplementary Fig. 7). However, we cannot exclude the possibility that the optical resolution is insufficient to resolve the exact topology of these higher-fold vertices, and these temporal structures represent aggregations of multiple closely positioned three-fold vertices (Supplementary Fig. 7). Future work with enhanced imaging resolution should be performed to more closely study how higher-fold vertices are formed at the onset of MCC integration.

Altogether, our results suggest that, after probing the neighboring tissue environment, MCCs induce junction collapse in a multi-step process (Fig. 5k). First, cells establish stable attachments with surrounding vertices. The integrating MCCs can then pull on the overlying junction and initiate its remodeling. Subsequently, myosin II accumulation at the junctions by the goblet cells reinforces junction collapse. Thus, integrating MCCs exert forces on a pliable environment to induce junction collapse and create advantageous insertion points.

## Myosin activity in the MCCs is required for junction remodeling and integration

We next explored how the integrating MCCs exert pulling forces to remodel the neighboring environment and produce favorable insertion points. A prime candidate for this would be myosin II, a key force-generating molecular motor[32], so we expressed the myosin sensor in the MCCs. We observed that myosin II localized to the leading edge of integrating cells, where it first appeared at the base of the filopodia (Fig. 6a, b, Supplementary Fig. 8a, b, and Supplementary Movie 14). We then quantified whether myosin II recruitment is sustained after the cells have inserted, and observed that myosin II was progressively enriched similarly to F-actin (Supplementary Fig. 8c, d). These data suggest that myosin II is recruited to the leading edge of the MCCs to facilitate cell integration. To test whether MCCs' ability to integrate depends on their capacity to exert forces on their neighbors, we assessed the impact of myosin II inhibition on MCCs integration. To this end, we mosaically expressed a constitutively active form of the myosin light chain phosphatase (CA-MYPT) in the MCCs, which dephosphorylates myosin II and negatively regulates actomyosin contractility[4] (Fig. 6c). Our dynamic imaging experiments verified that CA-MYPT expressing MCCs failed to both integrate and remodel the

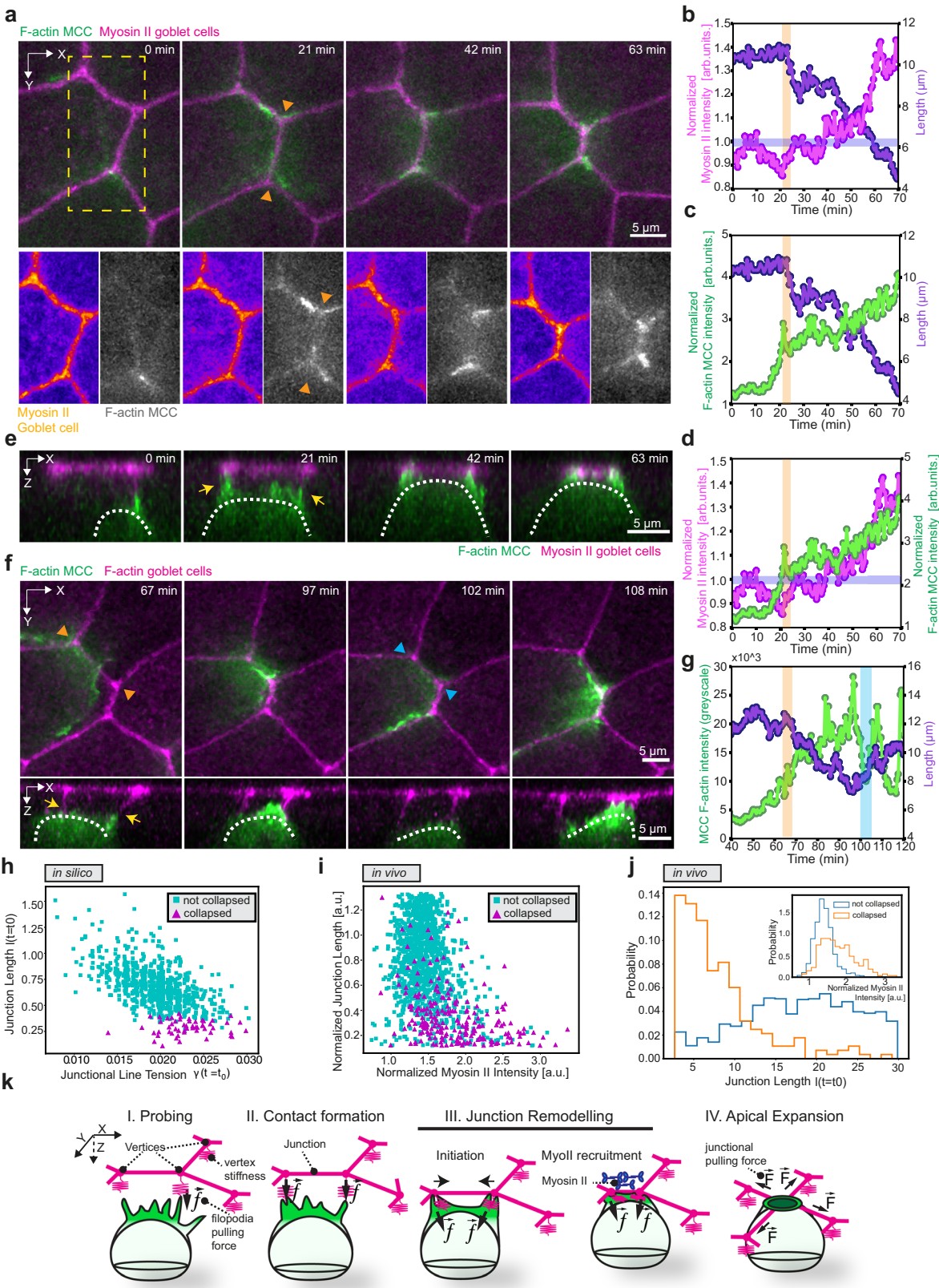

overlying junctions, despite extending filopodia that engage with the overlying vertices and positioning themselves at the epithelial vertices (Fig. 6c–f, Supplementary Fig. 8e, and Supplementary Movie 15). Together, the theoretical and experimental data demonstrate that MCCs initiate and then cooperatively drive the remodeling of epithelial junctions by pulling at the overlying vertices, which

facilitates the formation of the higher-fold vertices that favor epithelial integration.

## Discussion

Here, using radially integrating MCCs in vivo as a model system, we have begun to dissect the mechanisms regulating the mechanical

**Fig. 5 | MCCs remodel the overlying epithelium to enable cell integration. a, e** MCCs are labeled with α-tubulin::LifeAct-RFP (pseudo-colored in green in composite image and pseudo-colored in gray as separate channel) and myosin II in goblet cells is labeled by myosin II intrabody SF9-3xGFP (pseudo-colored in magenta in composite image and pseudo-colored in fire as separate channel). **a** Image sequence of epithelial junction remodeling during MCC integration. Scale bar: 5 μm. Orange arrowheads mark the onset of remodeling. **b** Normalized junctional myosin II intensity (magenta) and junction length (purple) during remodeling from **a**. Orange and blue bars mark the start of remodeling and normalized myosin II intensity equaling one, respectively. **c** Normalized MCC F-actin intensity (green) and junction length (purple) during remodeling from **a**. Orange bar marks the start of remodeling. **d** Normalized junctional myosin II (magenta) and normalized MCC F-actin (green) intensities during remodeling from **a**. Orange and blue bars mark the start of remodeling and normalized myosin II intensity equaling one, respectively. **e** Orthogonal (XZ) projections of **a**. Yellow arrows depict MCC contact with the epithelial vertices. Scale bar: 5 μm. **f** Vertex retraction during junction remodeling. MCC expresses α-tubulin::LifeAct-GFP (pseudo-colored in green) and goblet cells express nectin::utrophin-RFP (pseudo-colored in magenta). Orange arrowheads mark the onset of remodeling. Cyan arrowheads mark the loss of MCC contact with the vertices. Scale bar: 5 μm. White dotted line outlines the MCC contour. Yellow arrows depict contact with the epithelial vertices. **g** MCC F-actin fluorescence intensity (green) and junction length (purple) during junction collapse (indicated by the orange bar) and retraction (indicated by cyan bar) from **f**. **h, i** Stability-diagram of junction collapse in the normalized junction length-tension phase space from **h** in silico and **i** in vivo. The magenta triangles mark the junction length vs. line tension of collapsed junctions, while cyan squares represent non-collapsed junctions, when perturbed **h** by line tension changes (in silico) and **i** by MCC remodeling (in vivo). (See Supplementary Note 1). **j** Distribution of junction length and tension for collapsed and non-collapsed junctions in vivo showing stronger sensitivity of junction collapse to the initial length than to tension (shown in the inset). **k** Schematics representing the multi-step, cooperative process of junction remodeling.

interplay between an integrating cell and the surrounding tissue. Our experimental and theoretical results identify actin-based filopodia and epithelial vertices as the main players involved in probing tissue mechanics and put forward a concept of how a single cell can sense, interpret and remodel the neighboring cellular microenvironment to coordinate its behavior.

Our findings show that during radial intercalation, hundreds of migrating MCCs actively decide in which vertex to insert. This choice does not follow a straightforward "first-come-first-serve" principle. Cell integration is instead guided, as the migrating MCCs use filopodia to actively sense the vertices of the overlying epithelium. Therefore, we propose that filopodia provide the integrating MCCs with a parallel guiding mechanism to the recently described Scf/Kit biochemical signaling pathway[46]. While Scf/Kit controls both the spacing between neighboring MCCs and their overall affinity to the overlying epithelium, filopodia precisely inform the MCCs on the position of the vertices where they integrate into the tissue.

Our work further adds important regulators involved in cell integration. First, we reveal a previously uncharacterized localization of LSR to the tips of filopodia. Secondly, we find that LSR depletion in MCCs blocks filopodia formation, impairs cortical actin assembly and leads to integration defects. Finally, we show that myosin II is recruited to the base of filopodium and its downregulation leads to the impairment of MCCs integration, similarly to LSR knockdown. Altogether, our results describe the mechanistic basis for how filopodia are required for proper MCC integration within the overlying tissue and suggest that LSR might act as a potential mechanotransducer.

Importantly, our work addresses the fundamental question of how MCCs select a particular integration point. A recent study has described the propensity of MCCs toward higher-fold vertices, and that increasing a cell's capability to insert by promoting microtubule acetylation skews the integration propensity towards lower-fold vertices[23]. Nonetheless, why MCCs preferentially integrate at higher-fold vertices remains an open question. Our work addresses this unresolved phenomenon and proposes that a cell's decision on which vertex to integrate is ultimately determined by vertex stiffness. Interestingly, a similar mechanism of probing by filopodia is known to be used by migrating cells to measure temporal variations of local ECM stiffness in vitro[47]. We show that an out-of-plane pulling force can effectively probe vertex stiffness to identify an ideal vertex: one in which high enough line tension promotes vertex opening. Together, our results provide key insights into the basis of mechanical probing of tissues by filopodia in vivo.

Strikingly, our findings reveal that MCCs are able to remodel the overlying cell-cell junctions to form the higher-order vertices in which they predominantly integrate. Junction remodeling is known to be the driver of many morphological processes in epithelial tissues[48,49]. Until now, these processes have been largely characterized to be driven by

cellular forces generated within the epithelial layer[5]. However, our results show that integrating MCCs also drive remodeling to create the optimal mechanical environment for their integration within the tissue. This process relies on initial mechanical stimuli by integrating MCCs, which then cooperate with goblet cells to complete junction remodeling. While myosin II is required for cell movement through confined 3D environments[50,51], our data suggest that myosin impairment does not seem to induce any major changes in cell body displacement, and MCCs' ability to extend actin-based protrusions (Fig. 5c, e). Previous work has also shown that myosin inhibition impacts filopodia pulling, but not filopodia formation[32]. Altogether, we reason that myosin II inhibited MCCs fail to integrate due to their inability to exert pulling forces on their neighbors.

Beyond the direct implications of our findings in understanding the fundamental concepts of cell integration in vivo, the experimental and mathematical framework described here provides insights into how cells sense and transmit mechanical cues from their environment to the cytoskeletal machinery that ultimately drives cellular behavior, a deeply fundamental question in biology. Finally, as both filopodia and mechanical stimuli are involved in a plethora of processes, from development to cancer[52–54], we expect our work to help understand how these two essential players are intertwined to guide cell behavior in both normal and pathological conditions.

## Methods
The key Resources used in this publication are included in the Supplementary Table 1 of the Supplementary Information.

### *Xenopus laevis* embryo manipulation
*X. laevis* adult females were injected with 500 units/animal of Human Chorionic Gonadotropin (Chorulon[R]) to induce ovulation. Male frogs were sacrificed and their testis dissected for the sperm samples. *X. laevis* eggs were harvested, fertilized in vitro and dejellied with 3% cysteine (pH 7.9) solution after 2 h. Cleaving embryos were then washed and reared in 1/3× Marc's Modified Ringer's (MMR) solution. For mRNA and plasmid microinjection, embryos were transferred to a 2% Ficoll in 1/3× MMR solution and injected using glass needles and a universal micromanipulator. The Danish National Animal Ethics Committee reviewed and approved all animal procedures (Permit number 2017-15-0201-01237).

### Plasmid DNA/mRNA construct preparation
The primers used for cloning are listed in the Supplementary Table 1. Cloning of angulin-1/LSR into the pα-tubulin backbone was performed using a combination of the pENTRE™-dTOPO (Thermo Fisher Scientific) and Gateway™ systems. The LSR coding sequence was PCR-amplified and inserted into a pENTRE™ vector by an enzymatic reaction. The LSR CDS was then subcloned into a previously designed pα-

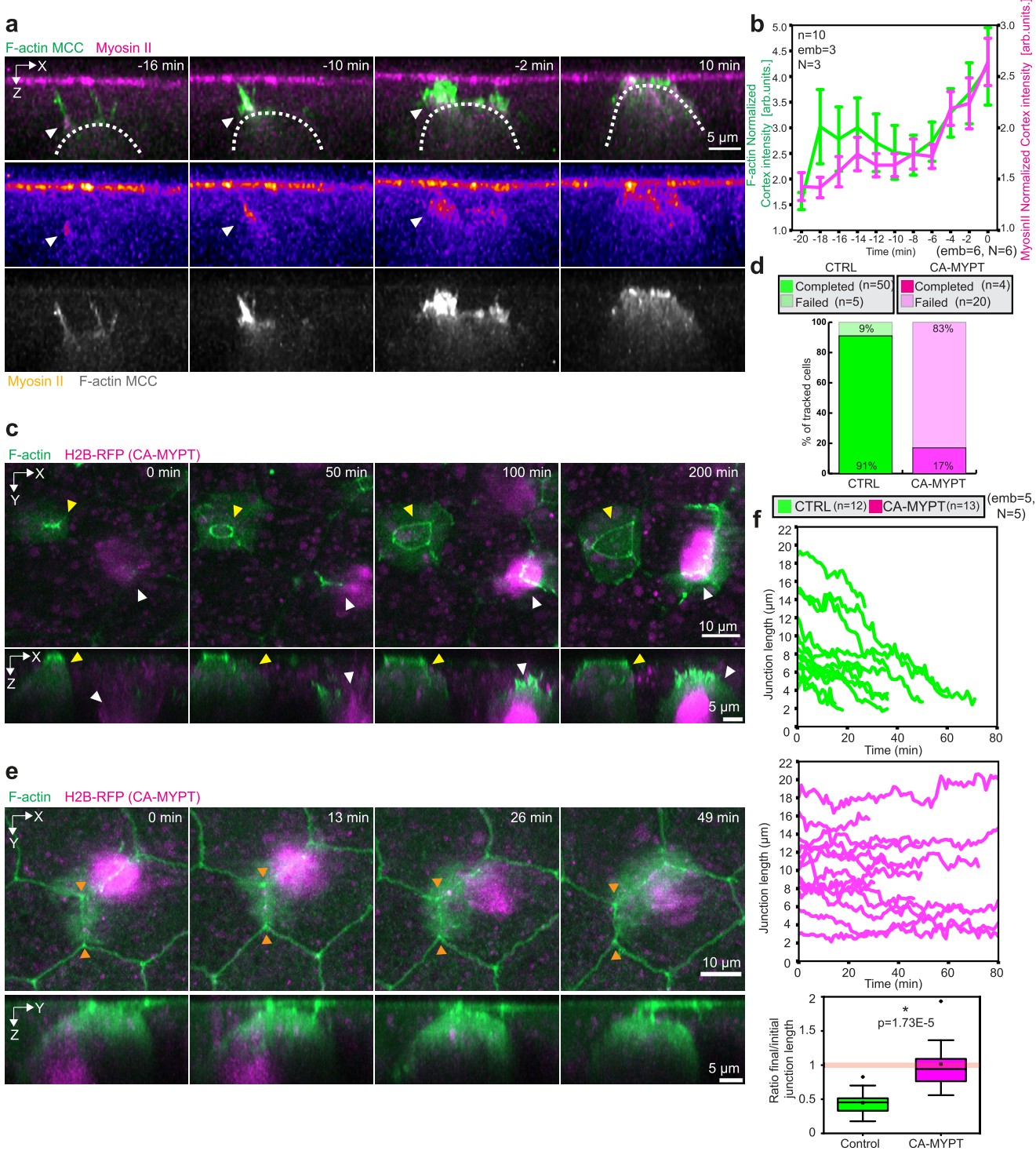

**Fig. 6 | Myosin II is specifically required in the MCCs for integration and epithelial junction remodeling. a** Orthogonal (XZ) projections of myosin II recruitment to the MCC leading edge during integration. MCC is labeled with α-tubulin::LifeAct-GFP (pseudo-colored in green in the composite image, pseudo-colored in gray as a separate channel) and myosin-II is labeled with the myosin intrabody SF9-3xGFP (pseudo-colored in magenta in the composite image, fire pseudo-colored as a separate channel). White arrowheads depict myosin recruitment. Scale bar: 5 μm. **b** Normalized myosin-II intensity (magenta) and normalized F-actin intensity (green) at the cortex of integrating MCCs. *T* = 0 marks the last tracked frame during integration (*n* = 10 cells from 3 embryos, *N* = 3 experiments). Data show mean ± SEM. **c** Image sequence of control MCC (yellow arrowhead) and CA-MYPT-overexpressing MCC (white arrowhead, marked with H2B-RFP pseudo-colored in magenta) during integration. Control and CA-MYPT overexpressing MCC express LifeAct-GFP mRNA (pseudo-colored in green). Scale bar: 10 μm.

**d** Quantification of MCC integration success rates for control and CA-MYPT overexpressing cells (nWT = 55 cells, nCA-MYPT = 24 cells from 5 embryos, *N* = 5 experiments). **e** Image sequence of a CAMYPT-overexpressing MCC attempting epithelial junction remodeling (orange arrowheads). Goblet cells and CA-MYPT overexpressing MCC express LifeAct-GFP mRNA (pseudo-colored in green). Scale bar: 5 μm. **f** Junction length tracking for control (green) and CA-MYPT-overexpressing MCCs (magenta) (nWT = 12 junctions, nCAMYPT = 13 junctions from 5 embryos, *N* = 5 experiments). Boxplots of final to initial junction length ratio in control and CA-MYPT MCCs. The orange line indicates no overall junctional remodeling (final to initial junction length ratio = 1). Boxes extend from the 25th to 75th percentiles, with a line at the median and whiskers representing range within 1.5 interquartile range. Two-tailed Mann–Whitney test with 0.05 significance level *p* = 1.73E−5.

tubulin by recombination using the Gateway™ LR clonase II enzyme mix (Thermo Fisher Scientific)[55]. Cloning of the SF9 myosin sensor into the pCS2+/3xGFP backbone was performed by restriction-ligation. Cloning of the LSR CDS into a pNectin backbone was performed by restriction-ligation. All plasmids were confirmed by restriction and sequencing (Eurofins Genomics).

## Construct synthesis and microinjection

pCS2+ plasmids were linearized with NotI-HF$^R$(NEB), purified by gel extraction and then used as templates for in vitro transcription. In vitro transcription of mRNAs was performed using the mMachine SP6 kit (Ambion). Synthesized mRNA was purified by LiCl precipitation. Plasmid and mRNA probes were microinjected together or separately into four-cell stage embryos with a single injection into each ventral blastomere. mRNA constructs were injected in 10 nl single injections as follows: LifeAct-GFP, 6 ng/µl; H2B-RFP, 20 ng/µl; LSR-3xGFP, 10 ng/µl; SF9-3xGFP, 8 ng/µl; CA-MYPT, 100 ng/µl. For MCC-specific labeling of the actin cortex, pα-tubulin LifeAct-GFP and pα-tubulin LifeAct-RFP were injected. For MCC-specific expression of LSR, pα-tubulin angulin-1-GFP was injected. For goblet cell-specific labeling of the actin cortex, pNectin Utrophin-RFP. For goblet cell-specific expression of LSR, pNectin LSR-GFP and pNectin LSR-GFP were injected. All plasmids were injected at 7.5 ng/µl. Unless specified, LSR labeling is performed by expressing LSR-3xGFP mRNA. For LSR knockdown experiments, 23.8 ng of LSR MO#1 and 40 ng of LSR MO#2 were injected (Gene Tools). For LSR MO and CA-MYPT experiments, the LSR MOs and CA-MYPT were co-injected with the fluorescent nuclear marker H2B-RFP at the 16- and 32-cell stages to mosaically target integrating MCCs or the superficial epithelium.

## Preparation of surface epithelium explants

Surface ectoderm epithelium explants (animal caps) were dissected from embryos at stage 8 of development. The explants were then placed on a fibronectin coated coverslip, cultured in Danilchik's for Amy explant culture media (DFA; 53 mM NaCl₂, 5 mM Na₂CO₃, 4.5 mM Potassium gluconate, 32 mM Sodium gluconate, 1 mM CaCl₂, 1 mM MgSO₄, 0,1% bovine serum albumin, pH 8.3) and kept in place with a coverslip. Prior to imaging, the explanted tissue was incubated 1h at RT, after which the coverslip was removed.

## Immunostaining

Immunostainings for LSR and pMyo were performed as previously described[35,56]. For LSR staining, LSR MO#1 injected embryos were fixed in 4% PFA in 1×PBS for 2h and then washed in 1×PBS for 20 minutes before blocking overnight at 4 °C in a blocking solution, 1×TBS with CasBlock™ (ThermoFisher). The rabbit anti-LSR antibody was used in a 1:50 dilution and incubated for two days at 4 °C. Embryos were then washed in blocking solution overnight and incubated with an anti-rabbit Alexa 647 secondary antibody (ThermoFisher) using a 1:250 dilution. After incubating for 24h, embryos were washed in the blocking solution overnight and then phalloidin-448 (Abberior) was added, according to lot number specification, for 2 h at room temperature. For pMyosin staining, embryos injected with SF9-3xGFP mRNA were fixed in 4% PFA in 1×PBS with 0.2% Triton X100 overnight, then washed in 1×PBS for 30 min before blocking for 2 h at 4 °C in a blocking solution (1×PBS with 0.2% Triton X100, 10% fetal bovine serum (FBS) and 1% bovine serum albumin (BSA)). The rabbit anti-phospho myosin antibody (Cell Signaling) was used in a 1:50 dilution and incubated overnight at 4 °C in a blocking solution. Embryos were then washed in 1×PBS with 0.1% Tween20 at 4 °C for 4 h and incubated overnight at 4 °C with an anti-rabbit Cy3 secondary antibody (Jackson Immunoresearch) using a 1:200 dilution and Alexa647 phalloidin (ThermoFisher), according to lot number specification. In both stainings, embryos were imaged immediately after finishing the protocol to guarantee staining quality.

## Confocal microscopy

All live imaging of early neurula-stage (Nieuwkoop and Faber, NF stage 13) *Xenopus* embryos was performed with confocal laser scanning inverted microscopes Zeiss LSM880 and LSM980 with Airyscan2 detector equipped with a ×40 C-Apochromat W autocorr M27 water immersion objective (NA = 1.2, working distance = 0.28 mm) or with a ×25 LD LCI Plan-Apochromat autocorr M27 water immersion objective (NA = 1.2, working distance = 0.28 mm)(Carl Zeiss Microscopy). Embryos had the vitelline membrane removed and were left to recover for 30 min before being mounted in a drop of 1% Ultra Low Melting Point Agarose (Sigma) prepared in 1/3× MMR. Embryos were then live imaged at room temperature. Movies of integrating MCCs were acquired with a ×40 objective with 30-, 60-, or 120-s intervals and 0.449 µm optical section in all figures except in Fig. 3 and Supplementary Fig. 5, where large-field of view movies of developing superficial epithelia were acquired with the ×25 objective at 300-s intervals. For imaging of surface ectoderm epithelium explants, time-lapse movies were acquired as described for developing embryos using a ×40 objective, with the following changes: images were acquired with a 2 µm optical section and a 1 min time interval. The time-lapse was run for approximately 16 h. The obtained tiles were stitched and Z sections were orthogonally projected using ZEN Black software (Carl Zeiss Microscopy). All movies were acquired in the regular confocal mode except in Supplementary Fig. 7, where the Airy Scan module was used for the superresolution imaging of rosette formation. Images were acquired using the Airy Scan module Super-resolution mode with a ×40 objective, 60-s intervals and 0.186 µm optical section. Images were deconvolved using the Airy Scan processing tool of ZEN Black software with standard settings (Carl Zeiss Microscopy).

## Laser ablations

To validate the SF9-3xGFP myosin sensor as a proxy for junction tension, we performed junction recoil measurements in stage 16 to stage 20 embryos expressing the SF9-3xGFP myosin II sensor. Laser ablations were performed using a 532nm pulse laser (>60 µJ pulses at 200Hz) at 20% power, with each ablation set for 5 ms. The ablation system was connected to a 3i spinning disk microscope with a Plan-Apochromat ×63 oil objective (N.A. = 1.4) mounted on an inverted Zeiss Axio Observer Z1 microscope (Marianas Imaging Workstation [3i −Intelligent Imaging Innovations]), equipped with a CSU-X1 spinning disk confocal head (Yokogawa) and an iXon Ultra 888 EM-CCD camera (Andor Technology). Vertex displacement was manually tracked using Fiji and the recoil velocities were calculated as previously described[18].

## Image processing and analysis

All image processing and analysis was performed using Fiji[57].

## Filopodia analysis pipeline and filopodia/cortex actin/myosin intensity analysis

The pipeline for the epithelial vertices and filopodia analysis was performed as follows. First, the 3D stacks of the MCC integration process were resliced (X–Z direction), and maximum projection was applied to all the slices at every time point. Then the vertices on either side of the cell contour were tracked using the manual tracking plugin (https://imagej.nih.gov/ij/plugins/track/track.html) in Fiji. Using the position coordinates, the distance between the left and right vertices was calculated. To mark the cell outline, the base of the filopodia was manually traced and the Region of Interests (ROIs) were recorded (Supplementary Fig. 1c). The cell tip given by the highest point of the cell outline and the vertex positions were used to calculate the distance between the cell tip and the vertices. Similarly, the cell tip and the center of the line connecting the left and right vertices were used to compute the distance between the cell and the epithelial surface. With the cell outline as a reference, the contour was shifted above and below with appropriate thickness to match the filopodia (ROI #3) and

the cell cortex (ROI #2) respectively (Supplementary Fig. 1c). Background intensity values were collected for correction (ROI#1)(Supplementary Fig. 1c). In order to identify the prominent filopodia, we normalized the mean intensity of every pixel along the filopodia contour with the mean intensity of cortex, and highlighted those pixels with the values above "1" in magenta (Fig. 1f, g). All the steps in this pipeline were performed using scripts written in Fiji (for data collection) and in MATLAB R2017b (for analysis and plotting). The first part of the pipeline was also adapted to determine the accumulation of F-actin and myosin at the cortex of integrating cells for Fig. 5b, Supplementary Fig. 4 and Fig. 8b, d. Different ROIs were extracted for quantitative analysis: 1−background, 2−trailing edge of the integrating cell, 3−cortex, and 4−filopodia of the integrating cell (see Supplementary Fig. 4d). Mean F-actin and myosin intensities at the leading edge (cortex or filopodia) were normalized to the respective mean intensities for the cell's trailing-edge. For Supplementary Fig. 4f, individual intensity measurements for each time point were pooled for statistical comparison of filopodia/cortex mean intensity between control and LSRMO#1 MCCs. All mean F-actin intensity measurements in MCCs were performed in cells expressing p-αtubulin::LifeAct-GFP except in Figure S4b, e, f, where LifeAct-GFP mRNA expressing cells were used. In Fig. 5b, $T = 0$ marks the last tracked frame (when MCC has reached the top of the superficial epithelial layer) during integration.

### 3D rendering of integrating MCCs

3D rendered images of integrating MCCs were obtained using the ClearVolume plugin from Fiji[58].

### Integration success quantification

Integration success was qualitatively defined by a cell's ability to expand its apical domain using the control cells as the reference. Control and depleted cells were pooled from the same embryos. Cell blockage, cell death and cell disappearance were manually quantified. Cells were quantified as blocked if they fail to expand their apical domains, as dying if they underwent apoptosis and fragmentation and as disappearing if they moved inwards from the superficial epithelial layer without undergoing obvious apoptosis and fragmentation (Supplementary Fig. 3a−d).

### Intensity quantifications of immunostained samples

To determine the efficiency of LSR depletion, we quantified LSR intensity at the epithelial vertices in LSR-immunostained mosaic stage-16 embryos with control and LSRMO#1 depleted epithelial cells (Supplementary Fig. 3e, f). In all, 3.5-μm diameter circular ROIs were manually drawn over the position of the vertices, which were determined using the F-actin channel. These ROIs were then used to collect the mean gray values for F-actin and LSR, which were normalized to the average F-actin and LSR intensity of control cells, respectively. Distributions of the normalized F-actin and LSR intensities for the control and LSR depleted cells were plotted using OriginPro 2020. To validate the SF9-3xGFP intrabody myosin sensor, we quantified the junctional intensity of phospho-myosinII and SF9-3xGFP in phospho-myosinII immunostained embryos (Supplementary Fig 5c). Junctions were manually segmented and mean gray values were then extracted for both channels and normalized to the average junctional intensity for each channel.

### Quantification of LSR accumulation in filopodia

LSR accumulation at the filopodia was manually quantified by drawing an ROI along the filopodium and using the Plot Profile tool in Fiji (Supplementary Fig. 2g). Mean gray values were then extracted for both channels and normalized to the mean intensity of the MCC's cytoplasmic region. Normalized intensity values were plotted from the base to the tip of the filopodium.

### Image segmentation and cell tracking

Sum intensity projection images of the SF9-3xGFP were segmented using the Cellpose segmentation algorithm[59]. We used the pre-trained model cyto2 with the following conditions: cell diameter 50, flow threshold 0 and cell probability threshold 6.0. Mistakes in the segmentation masks were then manually corrected using the Tissue Analyzer drawing function. Tissue Analyzer was used to track cells and junctions and to obtain the morphological information and intensity measurements on both[60]. Cells at the edges of the image that failed to be properly segmented were excluded from the subsequent analysis.

### Vertex-fold number, myosin II, and sum of line tensions quantification

The number of threefold and fourfold vertices was extracted using Tissue Analyzer and normalized to the total number of cells for each timepoint. Raw myosin intensity values at the epithelial junctions (using Maximum intensity projections of movies of embryos expressing the SF9-3xGFP myosin II sensor) were extracted using Tissue Analyzer[60]. Junctional myosin II mean intensity values were normalized to the average intensity of the two cells constituting the junction. Normalized mean junctional myosin II intensities are then used for the computation of the sum of line tensions by adding up the normalized junctional myosin II values of junctions constituting the vertex.

### Analysis of junction recoil velocities and pMyosin intensity

To estimate the trend of non-linear data, we performed quadratic polynomial regression. For each fit, we report the equation, goodness of fit ($R^2$) and $F$-test $p$ value. Myosin II values were subset for intensity >3. To compare the junctional tension between different fold vertices, data in Supplementary Fig. 5f was offset (below 6 μm) to remove impact of smaller junctions with higher recoil.

### Higher-fold integration, remodeling percentage, and junction length quantification

The segmented movies from the Tissue Analyzer were used to quantify the type of integration. Each integration was manually tracked to determine the local organization of the site of the expanding MCC and scored depending on the conformation of the neighboring cells (threefold, fourfold, fivefold, sixfold, or sevenfold/neighbors. Data from different embryos was manually aligned to a common reference set by the onset of MCC integration, which was defined as the 1% addition of new MCCs into the epithelium ($t = 0$ in Fig. 4c, f, h). Higher-fold integrations were then manually scored for the concurrence with the remodeling events. Remodeling events were scored by tracing back the original conformation of the vertices before MCC apical expansion. Only remodeling events simultaneous to MCC integration were scored as remodeling. From these events, the junction length distribution involved in forming higher-order vertices was calculated with $t = 0$ min marking the onset of MCC reaching the overlying junction. The other represented timepoints ($t = -20$ min, $t = +20$ min, $t = +40$ min, $t = +60$ min) were aligned to $t = 0$ min. Only events where one MCC can unequivocally be tracked throughout the remodeling event were quantified.

### Junction length and MCC myosin II quantification during junction remodeling

Junction length was manually tracked by tracing the junction outline using the segmented line function from Fiji on Maximum intensity projection images (exemplified in Supplementary Fig. 6a). Mean gray values were extracted from ROIs with 1 μm thickness. Junctional myosin II and F-actin mean intensities were normalized by dividing the mean intensity of a collapsing junction by the mean intensity of a non-dynamic junction (a junction that does not significantly change the length over time). To obtain the myosin and actin intensity at the leading edge of the MCC in Supplementary Fig. 8d, mean gray values

were extracted from ROIs with 1 μm thickness that traced the leading edge of the MCC. Myosin II and F-actin mean intensities were normalized by dividing the mean intensity of the MCCs leading edge by the mean intensity of a goblet cell's cytoplasmic region. Data were plotted using Plot2.

## Vertex pulling quantification

3D hyperstacks of MCC interacting with epithelial vertices were resliced ($X$–$Z$ direction) and projected to obtain a maximum intensity projection. The signal corresponding to LSR-3xGFP or LSR-GFP was filtered using a median filter with a 2-pixel kernel and then manually thresholded to obtain the epithelial vertex outline. Using the Analyze Particles tool of Fiji, we obtained ROIs outlining the vertex. The Feret's diameter (corresponding to the longest possible distance between two points in the ROI) was then used as a measure of vertex length (referred to as Feret length in the figures to avoid possible confusion). MCC F-actin mean intensity was measured inside the ROI as a proxy for the contact between MCC and the epithelial vertex and it was normalized to the mean F-actin intensity of the trailing edge of the cell. For average MCC F-actin intensity and Feret length, pulling events were aligned to the Feret length maximum ($T = 0$), depicting the peak of pulling at the vertex (Supplementary Fig. 2c).

## Statistics

Statistical analysis was performed using the Origin2020 software. Non-parametric Mann–Whitney $U$-tests were used for the analysis of statistical significance in Figs. 3i and 5f and Supplementary Figs. 3f, 4f, and 5f. One-way analysis of variance (ANOVA) with Tukey's test was used to compare different junction lengths with the reference junction length (junction length at the onset of MCC integration, $T = 0$) in Supplementary Fig. 5k. The experiments were not randomized, and no statistical method was used to select sample size. Result reproducibility was confirmed by performing independent experiments and all experiments have a minimum of three biological replicates from different clutches of eggs. Information on the statistical significance, number of cells/junctions ($n$), embryos (emb) and experiments ($N$) included in each panel is detailed in each figure, the corresponding figure legend, and Supplementary Table 2.

## Reporting summary

Further information on research design is available in the Nature Research Reporting Summary linked to this article.

# Data availability

The data and materials that support these findings are available within the article and its Supplementary Information files. Additional information and relevant raw data are available from the corresponding authors J.S. (jakub.sedzinski@sund.ku.dk) and A.D. (doostmohammadi@nbi.ku.dk) upon request. Source Data are provided with this paper.

# Code availability

The code used to analyze filopodia dynamics is available on the following github link (https://github.com/RaghavanThiagarajan/MCC_filopodia_quantification). The codes used for analyzing the remaining data in this work are available from the corresponding authors J.S. (jakub.sedzinski@sund.ku.dk) and A.D. (doostmohammadi@nbi.ku.dk) upon request.

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

## Acknowledgements

The authors thank all members of the Sedzinski lab and Active & Intelligent Matter Group members for comments and suggestions, Aritra Misra for performing the phospho-myosin stainings, Martin Proks for help in analyzing the laser ablation and phospho-myosin data, Girish Rajendraprasad and Marin Barisic for technical assistance and usage of the laser ablation system, Elke Ober and Mariaceleste Aragona for critical reading of the manuscript, Ann L. Miller, Edwin Munro and Elias H. Barriga for key reagents, and Benoit Aigouy for help with Tissue Analyzer. We acknowledge the Core Facility for Integrated Microscopy, Faculty of Health and Medical Sciences, University of Copenhagen and reNEW's microscopy specialist, Jutta M. Bulkescher, for training, technical expertise, support, and microscope use. J.S. acknowledges the support of the Novo Nordisk Foundation (grant No. NNF19OC0056962) and Leo Foundation (LF-OC-19-000219); A.D. acknowledges the support from the Novo Nordisk Foundation (grant No. NNF18SA0035142), Villum Fonden (Grant no. 29476), and funding from the European Union via ERC Starting Grant PhysCoMeT. A.A. acknowledges support from the Federal Ministry of Education and Research (Bundesministerium für Bildung und Forschung, BMBF) under project 031L0160. The Novo Nordisk Foundation Center for Stem Cell Biology and Novo Nordisk Foundation Center for Stem Cell Medicine are supported by Novo Nordisk Foundation grants (NNF17CC0027852 and NNF21CC0073729).

## Author contributions

G.V. performed all the experiments (except the immunostainings in Supplementary Fig. 5c and the live imaging of ectodermal explants in Supplementary Fig. 5a), analyzed the data and prepared the figures; M.T. helped with image segmentation, immunostainings and performed the live imaging of ectodermal explants in Supplementary Fig. 5a; R.T. helped with image analysis and developed the filopodia analysis pipeline; A.A. and A.D. developed the theoretical model and contributed to

data analysis and figures; J.S., A.D., G.V., and A.A. wrote the manuscript; All authors edited the manuscript; J.S. and G.V. conceived the project; J.S. wrote the first draft of the manuscript, acquired funding, and supervised the project.

## Competing interests

The authors declare no competing interests.
