## [Peer Review File · Nature Communications]

Multiciliated cells use filopodia to probe tissue mechanics during epithelial integration in vivoREVIEWER COMMENTS

Reviewer #1 (Remarks to the Author):

Ventura et al attempts to provide new in vivo insights on the contribution of mechanical signaling to cell integration. They propose that *Xenopus* multiciliated cell (MCC) precursors form filopodia to pull at the vertices of the overposing epithelia and that these filopodia probe vertices stiffness to identify the positions where MCC will integrate into the epithelial tissue. They also propose LSR as an important component of the mechanism mediating this “probe to integrate” mechanism. Then they make an effort to suggest that pulling forces are the driving motor behind MCCs remodelling of the epithelial junctions as these forces generate a permissive environment for integration. While I find these results of some relevance and that they could advance our understanding of the contribution of mechanical signaling during tissue morphogenesis, there are major points that require to be addressed before this work is published. See below.

Major points:

1. Although the authors are presenting the “Mechanics of cell integration in vivo” the study lacks direct measurements and mechanical perturbations of living tissues, methodologies which today are widely established in *Xenopus* (see references below). For instances, while SF9-3xGFP fluorescence was used as a proxy of tension when validating their model predictions, the conclusions would benefit of having laser ablations measurements as a much more direct mechanical readout of tension. Recoil speed analyses after laser ablation are often used in *Xenopus* and other species to estimate tension. Alternative, the authors can probe junction stiffness with AFM, a method that is also widely used in *Xenopus* (Koser et al 2016, Barriga et al 2018, Thompson et al 2019 and Shellard et al 2021, Moreira et al 2022, in press). Owing the superficial nature of MCC the apparent elastic properties of MCCs can be directly measured with these systems. Either approach could be applied as a control to validate the model and the current approach to line tension or could be extended to the parts of the article in which line tension has been used as a proxy of stiffness. These experiments would help to further validate the model presented by the authors and to experimentally strength the mechanical component of their work; an aspect which is missing in a work that aims to provide seminal steps to a mechanical understanding of cell intercalation.
2. The authors lack a control showing whether SF9-3xGFP fluorescence (developed for this study) correlates with phospho-myosin (an IF could help here). This could also help in Figure 5 to show that MYPT is maintaining myosin in a dephosphorylated state.

3. Along the text the authors provide orthogonal views of their imaging, but not in Fig5. Please add these images as it would help to support the conclusions of line 250 in which the authors state that CA-MYPT also impacted intercalation.

4. Statistical comparisons are made available for SF3f, but the experiment was not repeated 3 times, as stated by the authors. Please double check.

5. Statistical descriptions are missing in the figure legends. Difficult to understand what tests were used in each panel, if any, as comparisons are missing in some charts, particularly in line plots.

Minor points:

6. Abstract could be improved, i.e. not clear what the authors mean by homeostasis; cells do not integrate in disease? Also, very little was known about the interplay of migrating cells and their surroundings, but as per today there are several examples of it in the literature. Please update and clarify your abstract accordingly.

7. In Line 247 CA-MLCP is CA-MYPT.

8. In some sections the article is written by stating the will of the cells to execute an action. Please amend along the text.

Reviewer #2 (Remarks to the Author):

The paper entitled “Mechanics of cell integration in vivo” focuses on the role of cell-cell interaction during intercalation of MCCs in *Xenopus* epithelium.

Why is the paper titled like a review? A title should be used to give the main message of the study. As it stands it is misleading because, as a reviewer, part of the job is to assess whether the data presented match the message of the title. Can the authors claim this or that based of their data? Here there is no

claim. I am sure that they do not think their study explain mechanics of cell interaction as whole, so why such an odd and unhelpful choice?

The “mechanics” aspect of the paper is mostly inferred from descriptive data and simulations with little actual probing of forces/tensions/stiffness or functional assays attempting to validate their relevance.

While the authors provide a wealth of descriptive data supported by tons of high quality imaging the actual demonstration of most of their claims is lacking functional support and thus does not meet the requirement for publication in a high impact general publication such as Nature Communications. The story is better suited for a specialized journal. I would recommend that for doing so, the title be changed to match the main message of the story and most claims toned down to avoid misleading readers with statements that suggest experimental demonstration when only descriptive data are provided.

Specific comments:

The authors claim that “These F-actin-rich filopodia were dynamic and pointed at the cell junctions overlaying the MCCs”. Yes, but the videos also show plenty of filopodia that do not seem to point at cell junctions. Also, the way some data are presented makes it hard to interpret. I am not sure how to read graphs in panels 1g. This is supposed to show enrichment of filopodia at vertices. But I just do not understand the plots. Also, if that is the case, the authors did not seem to statistically test their hypothesis. They use these data to claim coordination between filopodia and MCC movement but it is merely the observation of a correlation not a demonstration of coordination between two events. It seems to me that such statement would only be reached by experimental perturbation of the system.

Then they go on by looking at LSR1 (aka Angulin1) and after a series of descriptive data claim that “Thus, our data demonstrate that MCCs use filopodia to directly interact with vertices by LSR-LSR mediated contacts.” However, only functional experiments can demonstrate that these cells indeed use LSR-LSR contacts. The fact that staining colocalize do not demonstrate functional relevance. The title of Figure 2 is completely misleading “LSR controls the interaction between MCCs and epithelial vertices”. This cannot be concluded from descriptive data.

From this point, they aim at targeting LSR by loss-of-function. For this they use an antisense Morpholino. Here I have a big issue. The sequence of the Morpholino is given however there are no mention of any controls anywhere in the main text or the supplementary data. It is not even stated whether it targets the ATG region or an exon-intron region. A loss-of-function, to be interpretable, needs to be validated for efficiency (is the tool leading to a knockdown of the target?) In their Ref30 an antibody against LSR1 was used so it is available to check for efficiency of their MO. Provided that this antibody actually

recognizes *Xenopus* LSR1 (is that an antibody against the *Xenopus* protein or an orthologue from another species. In the second case the Ab needs validation too). And more importantly, the tool needs to be checked for specificity by a rescue experiment. If not, one cannot distinguish between off-target effects and specific effects. This logic applies to any LOF (siRNA, CRISPR, MO). Also, LOF are usually confirmed by a different mean. A phenotype obtained by a validated MO should be compared with a phenotype generated by another independent validated method (dom-neg, CRISPR/gRNA at G0, at the very least a non-overlapping second MO either ATG or splicing). As it stands all data generated with LSR MO are no more than preliminary data and cannot be used in a published work. Therefore, all data, and associated conclusions, relying on this unvalidated MO should be dismissed: Figures S3 and S4.

The authors used a model to make prediction about local tension and stiffness, their relation with filopodia-based probing and the probability of intercalation. Then they go in vivo to test some of their predictions. The fact that they rely solely on Myosin-II distribution (again a descriptive dataset) to infer tension is strange. Why not laser ablation/recoil type of assay? To test whether the differential distribution of tension between the different types of junctions (3, 4, more cells) actually matches their prediction? It is unclear whether there is a technical hurdle here given the quality of their imaging, one would expect these type of assays to be possible. That would give actual relative tension data between the different types of junctions.

After that the authors looked at the interplay between MCC and goblet cells. Again numerous hypotheses and “conclusions” solely based on descriptive data. Looking carefully at the dynamics of a process is super interesting but is only a basis for designing functional assays. Descriptive data, as detailed as they are, only take you thus far. Some of these ideas are then put through their model where they find a correlation between junction collapse under specific length/tension situations. They then say that the experimental data are “qualitatively similar”??? Whatever that means. Aren’t statistical tests supposed to be used to assess agreement between datasets? Is “data look the same” supposed to mean that data are relevant?

Playing with the model they say that “the initial length of the junction plays a more dominant role in determining whether the junction collapses or not, as the order parameter is more sensitive to changes in the initial junction length”. One should be careful with this type of interpretation. Models are by nature a simplification and sensitivity to a given parameter may also be due to the fact that things are missing in the model. It does not indicate biological relevance or prevalence per se.

The main conclusion of these series of experiments, that MCCs actively pull on junctions and that this pulling actively triggers remodeling, has not been tested experimentally. It only stems from the interpretation of descriptive data and a theoretical model.

Next they eventually target myosin activity in MCCs. However, myosin is required for cell movement, deformation of the cell body etc. How can the authors be sure that the lack of intercalation is due to a lack of junction remodeling and not a failure of MCC cells to displace their cell body to a new location due to a cell-autonomous absence of MyosinII-based contractility? Is there anything in these datasets that preclude this interpretation of their inhibition assay?

Overall, the story is a very interesting example of cell-cell interaction during embryogenesis. However, most of the conclusions stem from descriptive work, not functional data and are thus still very speculative. Some of the experimental data rely on tools that were not validated either for efficiency or specificity. Finally, some data could be interpreted in a different way but authors do not balance their views with alternative explanations. For all these reasons, I do not recommend publication in Nature Communications.

Reviewer #3 (Remarks to the Author):

Ventura et al. investigated how multiciliated cell (MCC) precursors integrate into the superficial epithelial layer in the epidermis of *Xenopus* embryos. MCCs migrate below the epithelium before selecting an integration site and inserting at cell vertices in a process called radial intercalation. While recent studies have begun to shed light on this process, the underlying cellular mechanics was not understood and it was not clear how MCCs select their site of integration into the epithelium.

The authors present a very careful systematic analysis of the dynamic interplay between intercalating MCCs and the surrounding epithelial cells. They show that migrating MCCs extend dynamic filopodia that interact with epithelial cell vertices and thereby “probe” the local mechanical properties of the epithelium. Interestingly, the vertex-specific transmembrane protein LSR is enriched on tips of MCC filopodia as they pull on vertices, suggesting that LSR mediates interactions between filopodia and cell vertices. Consistent with this idea, the authors show that depletion of LSR from MCCs leads to impaired filopodial growth and reduced apical emergence of MCCs. They went on to test the idea that MCC filopodia are able to sense mechanical properties of the epithelium. They employed a theoretical model that simulates out-of-plane pulling forces at vertices by filopodia, yielding a map of local vertex “stiffness”. The model (the mathematical and physical basis of which I am not competent to judge) predicted that the line tension along cell-cell junctions constituting a vertex determines its propensity as a site for radial intercalation, with higher line tensions and resulting higher vertex stiffness favoring intercalation. Furthermore, heterogeneous line tension is predicted to promote the formation of higher-fold vertices where four or more cells meet, and such higher-fold vertices, as opposed to the more prevalent 3-fold vertices, are predicted to favor radial intercalation of MCCs. The authors tested these predictions using a combination of live imaging, cell labeling, and quantitative analyses. Consistent with model, they demonstrate that 4-fold vertices are indeed preferred over 3-fold vertices as sites for MCC insertion. Accumulation of myosin II (as detected using a GFP-labelled myosin II-specific nanobody) along cell-cell junctions is used as a proxy for line tension and vertex stiffness. Here, a more direct way

of assessing junctional tension, e.g. using laser ablations, would be necessary to validate the myosin II measurements and to substantiate the authors' conclusions regarding vertex mechanics, which are key to the study. Finally, the authors show that the interaction between integrating MCCs and surrounding epithelial cells drives junctional remodeling and the formation of higher-fold vertices, and that Myosin II activity is required in MCCs for these processes.

Altogether, this is an impressive body of very thorough work, combining live imaging and quantitative analyses with *in silico* modeling, the predictions of which, in turn, are tested by functional experiments. The results are convincing, carefully quantified and very clearly documented in text and figures. The work provides significant new insights into the mechanics of radial cell intercalation in a developmental context. These new findings are likely to have important implications also for other situations where cells move through tissues, such as transmigration of leukocytes or of metastatic cancer cells through endothelial vessel walls. Hence, the work presents a significant conceptual advance that will be of interest to a broad audience. The manuscript should be accepted for publication, given that the authors address the following points.

The title of the paper is extremely general and implies that an all-encompassing model for the mechanics of cell integration is presented. However, cell integration events in different tissues and between different cell types are likely to involve at least in part different mechanisms and mechanics (e.g., there may be cell integration events that do not take place at cell vertices). The authors should consider rephrasing the title accordingly.

The vertex-based model assumes that higher-order vertices represent direct contacts between 4 or more cells. However, higher-order vertices may in fact represent multiple closely spaced three-fold vertices (tricellular junctions) that cannot be resolved as separate entities by confocal microscopy. The authors should comment on this issue and its possible implications for MCC behavior. If cell vertices provide docking sites for MCC filopodia, could the presence of multiple closely adjacent docking sites explain why such apparent higher-order vertices provide preferential sites for MCC integration?

The use of the anti-myosin II nanobody (SF9-3xGFP) as a measure for junctional tension needs to be validated. What does the SF9 nanobody recognize? Does it interfere with myosin II function? How does the distribution of SF9-3xGFP signals correspond to the distribution of active myosin II (phospho-myosin II)? This should be straightforward to address by immunostainings.

Although the idea that homotypic LSR-LSR contacts mediate interactions between MCC filopodia and cell vertices is persuasive, no evidence for a direct LSR-mediated interaction (as claimed by the authors) is provided. Does LSR mediate homophilic cell adhesion? In the absence of such evidence, the authors need to tone-down their statements that "... our data demonstrate that MCCs use filopodia to directly interact with vertices by LSR-LSR mediated contacts" (line 110 and elsewhere).

Fig. 1c: the color code is confusing. A more intuitive continuous (unidirectional) color scale should be used.

Fig. 3e is not showing a probability density function (PDF). $K\delta$ is a continuous (random) variable. As such, the integral of the PDF must equal 1. This is not the case in the presented graph. The authors should use a correct way to estimate the PDF of $K\delta$, for instance by using kernel density estimations. Alternatively, a simple histogram could be shown.

Fig. S3f: The graphs are lacking a legend on the Y-axis.

We thank the reviewers for the helpful and enthusiastic comments on this paper, which we are certain greatly contributed to the improved quality of this revised version. Combined with the reviewers' suggestions, we have clarified some of our work's key points and made important changes to the language used to describe the process of cell integration. Altogether, we think these changes have further strengthened our work's message. Below, we present an itemized list of changes made in response to these comments, with changes to the manuscript marked in red.

Reviewer #1 (Remarks to the Author):

Ventura et al attempts to provide new in vivo insights on the contribution of mechanical signaling to cell integration. They propose that Xenopus multiciliated cell (MCC) precursors form filopodia to pull at the vertices of the overposing epithelia and that these filopodia probe vertices stiffness to identify the positions where MCC will integrate into the epithelial tissue. They also propose LSR as an important component of the mechanism mediating this “probe to integrate” mechanism. Then they make an effort to suggest that pulling forces are the driving motor behind MCCs remodelling of the epithelial junctions as these forces generate a permissive environment for integration. While I find these results of some relevance and that they could advance our understanding of the contribution of mechanical signaling during tissue morphogenesis, there are major points that require to be addressed before this work is published. See below.

Major points:

1. Although the authors are presenting the “Mechanics of cell integration in vivo” the study lacks direct measurements and mechanical perturbations of living tissues, methodologies which today are widely established in Xenopus (see references below). For instances, while SF9-3xGFP fluorescence was used as a proxy of tension when validating their model predictions, the conclusions would benefit of having laser ablations measurements as a much more direct mechanical readout of tension. Recoil speed analyses after laser ablation are often used in Xenopus and other species to estimate tension. Alternative, the authors can probe junction stiffness with AFM, a method that is also widely used in Xenopus (Koser et al 2016, Barriga et al 2018, Thompson et al 2019 and Shellard et al 2021, Moreira et al 2022, in press). Owing the superficial nature of MCC the apparent elastic properties of MCCs can be directly measured with these systems. Either approach could be applied as a control to validate the model and the current approach to line tension or could be extended to the parts of the article in which line tension has been used as a proxy of stiffness. These

experiments would help to further validate the model presented by the authors and to experimentally strength the mechanical component of their work; an aspect which is missing in a work that aims to provide seminal steps to a mechanical understanding of cell intercalation.

We agree that providing a direct mechanical readout of tension is essential, and we regret not including it in the original manuscript. We have now added to the revised manuscript laser ablation data that support our use of the SF9-3xGFP myosin-II sensor as a proxy for line tension (included in Supplementary Fig. 5). Our data show that both recoil and myosin-II intensity scale with junction length, with shorter junctions showing increased levels of myosin-II intensity and recoil velocities upon ablation. Moreover, using laser ablations, we show no significant difference in junction tension between junctions connected to 3-fold- and 4-fold-vertices, which is consistent with our theoretical model's prediction that increased vertex stiffness in higher-fold vertices is due to the higher number of connecting junctions.

2. The authors lack a control showing whether SF9-3xGFP fluorescence (developed for this study) correlates with phospho-myosin (an IF could help here). This could also help in Figure 5 to show that MYPT is maintaining myosin in a dephosphorylated state.

We have now included in the revised manuscript data on our SF9-3xGFP sensor and the corresponding phospho-myosin (pMyosin-II) immunostainings (Supplementary Figure 5c and 5e). The phospho-myosin stainings are consistent with both our *in vivo* recoil measurements and our SF9-3xGFP fluorescent intensity measurements, thus validating the use of the SF9-3xGFP as a myosin-II sensor. In addition, MYPT inhibition of myosin-II activity has been previously documented (Weiser et al., 2009) and successfully used to decrease myosin activity and tissue tension in *Xenopus* (Barriga et al. 2018; Yamamoto et al. 2021; Marchant et al. 2021).

3. Along the text the authors provide orthogonal views of their imaging, but not in Fig5. Please add these images as it would help to support the conclusions of line 250 in which the authors state that CA-MYPT also impacted intercalation.

We have now included the orthogonal views in Fig. 5c and 5e, which show that MCCs expressing CA-MYPT extend actin filopodia and interact with overlying vertices but fail to remodel the neighboring junctions and integrate within the superficial epithelium.

4. Statistical comparisons are made available for SF3f, but the experiment was not repeated 3 times, as stated by the authors. Please double check.

We have now corrected this mistake.

5. Statistical descriptions are missing in the figure legends. Difficult to understand what tests were used in each panel, if any, as comparisons are missing in some charts, particularly in line plots.

We have now added new statistical comparisons, together with a description of the statistical tests used in Fig.3, Fig.5, Supplementary Fig.3, Supplementary Fig.4 and Supplementary Fig.5 in the respective figure legends.

Minor points:

6. Abstract could be improved, i.e. not clear what the authors mean by homeostasis; cells do not integrate in disease? Also, very little was known about the interplay of migrating cells and their surroundings, but as per today there are several examples of it in the literature. Please update and clarify your abstract accordingly.

This is a good point, and we have now clarified the abstract, emphasizing the need to understand the mechanical interplay between migrating cells and the surrounding tissue environment.

7. In Line 247 CA-MLCP is CA-MYPT.

We have now corrected this typo.

8. In some sections the article is written by stating the will of the cells to execute an action. Please amend along the text.

We and others have read the manuscript multiple times, and it is unclear which particular passages should be amended. However, if the reviewer is referring to the cell decision-making process, we do not assume that cells are conscious entities but rather apply concepts from the mechanosensing field in which the integrating cell probes and interprets mechanical cues from the surrounding environment to control its behavior.

Reviewer #2 (Remarks to the Author):

1. The paper entitled “Mechanics of cell integration *in vivo*” focuses on the role of cell-cell interaction during intercalation of MCCs in *Xenopus* epithelium.

Why is the paper titled like a review? A title should be used to give the main message of the study. As it stands it is misleading because, as a reviewer, part of the job is to assess whether the data presented match the message of the title. Can the authors claim this or that based of their data? Here there is no claim. I am sure that they do not think their study explain mechanics of cell interaction as whole, so why such an odd and unhelpful choice?

We acknowledge the referee’s point and have had similar comments from the other referees. We have now changed the title to describe the study’s message more accurately. The title reads: **"Multiciliated cells use filopodia to probe tissue mechanics during epithelial integration *in vivo*"**

2. The “mechanics” aspect of the paper is mostly inferred from descriptive data and simulations with little actual probing of forces/tensions/stiffness or functional assays attempting to validate their relevance.

We have now included laser ablation data that supports the use of the myosin-II probe as a tension sensor (included in Supplementary Fig. 5). Our data show that both recoil and myosin-II intensity scale with junction length, with shorter junctions showing increased levels of myosin-II intensity and recoil velocities upon ablation. This is also corroborated by our stainings for active myosin II with phosphomyosin II specific antibody. We would like to emphasize that two different ways of probing tissue mechanics *in vivo*, i.e., through myosin concentration and laser ablation, have confirmed the predicted vertex stiffness patterns by the *in silico* model. As such, we believe that with the additional data we have now provided, the mechanics aspect of the paper has been further improved.

3. While the authors provide a wealth of descriptive data supported by tons of high quality imaging the actual demonstration of most of their claims is lacking functional support and thus does not meet the requirement for publication in a high impact general publication such as *Nature Communications*. The story is be better suited for a specialized journal. I would recommend that for doing so, the title be changed to match the main message of the story and most claims toned down to avoid misleading readers with statements that suggest experimental demonstration when only descriptive data are provided.

We respectfully dispute the assertion that our paper is only descriptive. We provide several lines of quantitative evidence for newly discovered phenomena, including the mechanical interaction between the MCCs and their neighboring epithelial cells through the epithelial vertices, with LSR and myosin II playing a central role in this process, identifying the mechanically susceptible points in the tissue for MCC integration, and revealing how such mechanical susceptible points are generated by the MCCs. Importantly, we support these novel findings both by molecular perturbations (LSR knockdown and overexpression of constitutively active form of a myosin phosphatase (CA-MYPT)) and theoretical modeling. As discussed in the point-by-point reply to the specific comments below, we have complemented our analyses with additional tension measurements by laser ablations, further strengthening our study.

Specific comments:

3a. The authors claim that “These F-actin-rich filopodia were dynamic and pointed at the cell junctions overlaying the MCCs”. Yes, but the videos also show plenty of filopodia that do not seem to point at cell junctions. Also, the way some data are presented makes it hard to interpret. I am not sure how to read graphs in panels 1g. This is supposed to show enrichment of filopodia at vertices. But I just do not understand the plots. Also, if that is the case, the authors did not seem to statistically test their hypothesis. They use these data to claim coordination between filopodia and MCC movement but it is merely the observation of a correlation not a demonstration of coordination between two events. It seems to me that such statement would only be reached by experimental perturbation of the system.

We wish to apologize to the reviewer here, as we realized that these complex graphical representations might be difficult to understand. We have now changed the text to provide a clearer, more succinct explanation of the analysis. This is supported by the Methods section and by Supplementary Fig.1 and Supplementary Video 2 which visually describe the methodology. We have now included in Supplementary Fig. 1 a representation of how the plots can be read.

We agree with the reviewer that we have only observed a correlation between filopodia dynamics and MCC movement. In the original manuscript, we included these data to strengthen the point about the relationship between vertices and filopodia dynamics. However, upon the reviewer's comment, we note that it is a very complex issue to untangle filopodia dynamics and cell body movement, as integrating MCC can interact with several TCJs at any given moment in a complex 3D environment. Thus, we have removed any mention that filopodia direct MCC movement from the main text together

with Supplementary Fig.1e and Supplementary Fig.1d. We have also changed the titles of the figures for Fig.1 (before: Filopodia guide radial intercalation of multiciliated cells (MCCs)., now: **Multiciliated cells probe the neighboring environment during integration**) and Supplementary Fig.1 (before: Vertex probing during MCC radial intercalation., now: **Filopodia dynamics during the probing phase of MCC integration.**).

We would, however, like to emphasize that the main message of these graphs is that MCCs use filopodia to interact with the vertices of their neighboring goblet cells. From our live imaging data it is apparent that the MCCs use filopodia to interact with the neighboring cells. To further support this, we have developed an image analysis pipeline to better understand the relationship between the protrusions extended by the MCCs and the epithelial vertices, which confirms that MCCs intimately interact with the neighboring epithelial vertices by using filopodia. The enrichment of the filopodia at the vertices, shown in Figure 1g, is a qualitative statement based on the visualization of the filopodia position relative to the position of the vertices. Quantification of the enrichment of filopodia at vertices is a complex image processing problem that would require either formulation of an arbitrary measure of an enrichment that could be applied to the 2D representation of filopodia dynamics maps (Fig. 1g) or would need to segment the filopodia in 3D, which as pointed above, is extremely challenging. Thus, we feel that a qualitative visualization of filopodia enrichment is, in this case, far more informative and further quantitative analysis of the filopodia enrichment will not enhance the otherwise highly quantitative story arc. We hope that the reviewers and the Editor can be convinced to agree.

3b. Then they go on by looking at LSR1 (aka Angulin1) and after a series of descriptive data claim that “Thus, our data demonstrate that MCCs use filopodia to directly interact with vertices by LSR-LSR mediated contacts.” However, only functional experiments can demonstrate that these cells indeed use LSR-LSR contacts. The fact that staining colocalize do not demonstrate functional relevance. The title of Figure 2 is completely misleading “LSR controls the interaction between MCCs and epithelial vertices”. This cannot be concluded from descriptive data.

We agree that it is essential to show functional data to claim the existence of direct LSR-LSR interactions. We now changed this statement and the title of Figure 2 to reflect LSR's role in MCC integration (before: LSR controls the interaction between MCCs and epithelial vertices., now: **Integrating MCCs pull on the epithelial vertices.**).

3c. From this point, they aim at targeting LSR by loss-of-function. For this they use an antisense Morpholino. Here I have a big issue. The sequence of the Morpholino is given however there are no mention of any controls anywhere in the main text or the

supplementary data. It is not even stated whether it targets the ATG region or an exon-intron region. A loss-of-function, to be interpretable, needs to be validated for efficiency (is the tool leading to a knockdown of the target?) In their Ref30 an antibody against LSR1 was used so it is available to check for efficiency of their MO. Provided that this antibody actually recognizes Xenopus LSR1 (is that an antibody against the Xenopus protein or an orthologue from another species. In the second case the Ab needs validation too). And more importantly, the tool needs to be checked for specificity by a rescue experiment. If not, one cannot distinguish between off-target effects and specific effects. This logic applies to any LOF (siRNA, CRISPR, MO).

We thank the reviewer for their suggestions, as we had missed important information about the LSR MO used to down-regulate LSR activity and the antibody used to validate it. We have now included this information in the Materials and Methods section. In the first submitted version of the paper, we used a translation blocking MO (targeting the translation start ATG region), which we have now listed as LSR MO#1 to distinguish it from the second splice-blocking LSR MO (listed now as LSR MO#2). To validate the LSR MO#1, we performed immunostainings with an antibody specifically designed against *Xenopus laevis* LSR, which we obtained from Prof. Ann L. Miller's Lab (Higashi et al., 2016). The immunostainings and the fluorescent intensity quantifications have been included in Supplementary Fig. 3e and f. The stainings with the LSR antibody were performed in mosaic embryos where patches of LSR-depleted epithelial cells are neighbored by control epithelial cells, which allows us to compare the efficiency of the depletion (Supplementary Fig. 3e). In these stainings, we observe that LSR MO#1 injections effectively knockdown LSR function, as the LSR signal is lost at the epithelial vertices (Supplementary Fig.3e). We have confirmed this by quantifying LSR intensity at TCJs and showing that it is significantly decreased (Supplementary Fig.3f).

3d. Also, LOF are usually confirmed by a different mean. A phenotype obtained by a validated MO should be compared with a phenotype generated by another independent validated method (dom-neg, CRISPR/gRNA at G0, at the very least a non-overlapping second MO either ATG or splicing). As it stands all data generated with LSR MO are no more than preliminary data and cannot be used in a published work. Therefore, all data, and associated conclusions, relying on this unvalidated MO should be dismissed: Figures S3 and S4.

We have now validated the LSR MO#1 using a splice-blocking LSR MO (LSR MO#2). We observed a consistent phenotype with the translation blocking LSR MO, with MCCs' showing a similar decrease ability to integrate. We have now included in Supplementary Fig. 3g and h. We would also like to point out that LSR depletion impacts cortical F-actin in epithelial cells (Supplementary Fig. 3f), which is in line with previous work that

describes how tricellular junction (TCJ) proteins regulate actin dynamics in epithelia. LSR is known to be responsible for recruiting Tricellulin to the TCJs/ epithelial vertices, which in turn recruits the actin regulators Tuba and Cdc42 to the TCJs (Higashi et al. 2013; Oda et al. 2014). This, combined with data showing that LSR depletion in the MCCs blocks filopodia formation and greatly reduces cortical actin, that LSR localizes to filopodia and that LSR overexpression induces the formation of ectopic filopodia in integrating MCC, suggests a general function for LSR in regulating actin dynamics. We have also tried to perform rescuing experiments but regret that we could not rescue the morphant phenotype by overexpressing LSR-3xGFP. This is, however, not uncommon in *Xenopus*, especially for the late-stage phenotypes. Critically, we used two different morpholinos targeting distinct regions of the LSR transcript and obtained consistent phenotypes - impairment of filopodia activity and cell integration defects. We hope that the reviewers will be convinced that this experiment is adequately controlled, and we are confident that these additional experiments establish a key functional role for LSR during cell integration.

3e. The authors used a model to make prediction about local tension and stiffness, their relation with filopodia-based probing and the probability of intercalation. Then they go in vivo to test some of their predictions. The fact that they rely solely on Myosin-II distribution (again a descriptive dataset) to infer tension is strange. Why not laser ablation/recoil type of assay? To test whether the differential distribution of tension between the different types of junctions (3, 4, more cells) actually matches their prediction? It is unclear whether there is a technical hurdle here given the quality of their imaging, one would expect these type of assays to be possible. That would give actual relative tension data between the different types of junctions.

We agree with the reviewer that the direct methodology to infer junctional tension would strengthen the model predictions. We have now performed laser ablation of epithelial junctions and calculated their recoil to infer junctional tension. These results have been included in Supplementary Fig. 5d and f and support our use of the SF9-3xGFP myosin-II sensor, while further validating our theoretical model. We have also performed immunostainings for phospho-myosin II to strengthen the validity of the myosin-II sensor as a readout of junctional tension (Supplementary Fig. 5c and e).

We failed to make ourselves clear and would like to emphasize that our model does not predict that junctional tension is different between 3-fold and higher-fold vertices, as indicated by the reviewer. Rather, our model predicts that the vertex stiffness of higher-fold vertices is greater than the 3-fold vertices. Since vertex stiffness depends on the sum of the line tensions constituting a vertex, our model predicts that higher-fold

vertices would open easier compared to lower-fold vertices. To put it simply, the 4-fold vertex (built by 4 pulling junctions) would open easier compared to the 3-fold vertex (3 pulling junctions). The model does not predict that junctions of 4-fold vertices are under more tension than junctions of 3-fold vertices. We consider the heterogeneous line tension in the model because it is energetically required to form the higher-fold vertices we observed *in vivo*. Without this assumption, there will be only 3-fold vertices having exactly the same junctional tensions.

When we performed laser ablations of junctions connected to either 4-fold and 3-fold vertices, we found no statistical difference in the junctional tension between the two (Supplementary Fig. 5f). This supports the model prediction that the number of junctions is the predominant factor that determines vertex stiffness and consequently the probability of vertex opening.

We have also extended the model description in the main text, under the "MCCs probe local vertex stiffness in the overlying epithelium" section, to better explain the concept of vertex stiffness and the probability of vertex opening.

3f. After that the authors looked at the interplay between MCC and goblet cells. Again numerous hypotheses and "conclusions" solely based on descriptive data. Looking carefully at the dynamics of a process is super interesting but is only a basis for designing functional assays. Descriptive data, as detailed as they are, only take you thus far. Some of these ideas are then put through their model where they find a correlation between junction collapse under specific length/tension situations. They then say that the experimental data are "qualitatively similar"??? Whatever that means. Aren't statistical tests supposed to be used to assess agreement between datasets? Is "data look the same" supposed to mean that data are relevant? Playing with the model they say that "the initial length of the junction plays a more dominant role in determining whether the junction collapses or not, as the order parameter is more sensitive to changes in the initial junction length". One should be careful with this type of interpretation. Models are by nature a simplification and sensitivity to a given parameter may also be due to the fact that things are missing in the model. It does not indicate biological relevance or prevalence per se.

The reviewer here is referring to this particular paragraph: "*In order to quantify the junction fate under tensional perturbation, we define l_f/l_i as the order parameter characterizing the ratio of the particular junction's length after applying a positive perturbation to its initial junctional tension (Fig. 4h). The simulation results showed that 4-fold vertices were formed from the collapse of junctions with sufficiently short length and sufficiently large tension (Fig. 4h). A qualitatively similar phase diagram is observed*

for the experimental data (Fig. 4i). Moreover, compared to the line tension, the initial length of the junction plays a more dominant role in determining whether the junction collapses or not, as the order parameter is more sensitive to changes in the initial junction length (Fig. 4h-j).” We have now changed this passage to: **“In order to quantify the junction fate under tensional perturbation, we define l_f/l_i as the order parameter characterizing the ratio of the particular junction's length after applying a positive perturbation to its initial junctional tension (Fig. 4h). The simulation results showed that 4-fold vertices were formed by the collapse of junctions with sufficiently short length and sufficiently large tension (Fig. 4h). Experimental data showed a similar trend for the collapse of junctions with varying junction length and tension (Fig. 4i). Moreover, compared to the line tension, the initial length of the junction plays a more dominant role in determining whether a junction collapses or not, as the probability of junction collapse is more sensitive to changes in the initial junction length (Fig. 4h-j).”**

In addition, we note the reviewer's comment about theoretical models being a simplification and sensitive to a given parameter. Here we have proposed a minimal biophysical model that accounts for the mechanical properties of the superficial epithelium and the interaction between epithelial and integrating cells. We show that these minimal (but essential) ingredients are sufficient to capture several key features of the mechanics of this particular tissue. The specific feature that is discussed in the highlighted sentence is that the weak spots (shorter bonds) in the epithelium can be inferred from the network geometry. This is helpful because we did not need to change any parameters that define the model to see this experimentally observed feature. Thus, the biological phenomena we observed *in vivo* can be seen *in silico* as a consequence of only accounting for key mechanical interactions. Consequently, while the reviewer is correct that models are a simplification of a biological process, the fact that a minimal number of parameters used in our model accurately predicts the experimentally-verified observations does indeed indicate that our model has strong predictive power.

3g. The main conclusion of these series of experiments, that MCCs actively pull on junctions and that this pulling actively triggers remodeling, has not been tested experimentally. It only stems from the interpretation of descriptive data and a theoretical model.

We respectfully disagree with this assertion. We effectively tested the prediction that MCCs pull on junctions, which then triggers remodeling, by downregulating myosin II activity specifically in the MCCs. In Fig. 5e and f, we describe how the inhibition of myosin II activity in the MCCs effectively blocks junction remodeling by MCCs. The referee indeed acknowledges this very point in the following comment.

3h. Next they eventually target myosin activity in MCCs. However, myosin is required for cell movement, deformation of the cell body etc. How can the authors be sure that the lack of intercalation is due to a lack of junction remodeling and not a failure of MCC cells to displace their cell body to a new location due to a cell-autonomous absence of MyosinII-based contractility? Is there anything in there datasets that preclude this interpretation of their inhibition assay?

A more exhaustive characterization of the CA-MYPT expressing MCCs will undoubtedly provide exciting results, but we respectfully argue that such work is beyond the scope of the present paper for reasons both conceptual and practical.

Conceptually, we think that a more comprehensive study of the effect of myosin II inhibition will not enhance the results of this manuscript. While we cannot be completely certain that myosin inhibition impacts other processes other than junction remodeling and filopodia pulling, our data blocking myosin activity in the MCCs suggests that other critical processes, such as cell movement and protrusion formation, remain unaffected: CA-MYPT expression does not seem to induce any major changes in cell body displacement and MCCs' ability to deform, and CA-MYPT expressing cells are still able to extend actin-based protrusions and position themselves at the epithelial vertices (Fig. 5c,e). We have now included these observations in the manuscript. Similarly, work in cell culture has shown that while Myosin II inhibition impacts filopodia ability to pull on the fibronectin matrix, it does not impact filopodia formation (Alieva et al., 2019). We feel that further dissection of the myosin II-based contractility in the integrating MCCs would actually distract the reader from what is already a complex narrative. We hope the reviewers and the Editor can be convinced to agree.

Practically, fully dissecting the different roles myosin-II plays in MCC integration is far less straightforward than it seems. We would have to perform simultaneously temporal and spatially controlled inhibition of myosin-II. While this is theoretically possible using an optogenetic system for myosin inhibition (as the OptoGEF system (Valon et al., 2016)), it would be extremely time-consuming and challenging to implement considering the limitations of our system (three transgenic constructs would need to be injected in a mosaic fashion to create tissue where only the integrating MCCs express the optoGEF system). Moreover, the OptoGEF system is not highly efficient in the *Xenopus* mucociliary epithelium (personal communication). Thus, we feel those experiments are beyond the scope of the current work.

3i. Overall, the story is a very interesting example of cell-cell interaction during embryogenesis. However, most of the conclusions stem from descriptive work, not functional data and are thus still very speculative. Some of the experimental data rely on

tools that were not validated either for efficiency of specificity. Finally, some data could be interpreted in a different way but authors do not balance their views with alternative explanations. For all these reasons, I do not recommend publication in Nature Communcaitions.

We are grateful that the referee assessed our findings as “a very interesting example of cell-cell interaction during embryogenesis”. We are confident that our extensive additional work described above has significantly strengthened the main conclusions, while we have also changed parts of the main text to provide a more balanced view of the results (such as the reviewer’s pertinent observation on myosin inhibition). Below we highlight multiple lines of detailed analysis that definitely distinguish our work from being purely descriptive and speculative:

We have characterized a novel phenomenon in which migrating cells probe tissue mechanics by pulling at the vertices of the overlying epithelium. We used mathematical modeling to analytically dissect this complex process. We have tested the model’s predictions with thorough quantifications and confirmed the applicability of the myosin-II sensor as a readout for junctional tension by phospho-myosin stainings and junctional laser ablations.

We identified LSR and myosin II as important molecular players regulating the process of cell integration and provided functional data to back up the conclusions.

We showed for the first time that integrating cells can remodel their neighboring epithelium, providing a novel perspective on how individual cells are capable of remodeling their environment to the tissue mechanics field. In addition, we describe how the pulling force used for probing and remodeling is generated by perturbing myosin-II activity specifically within the integrating cells.

Reviewer #3 (Remarks to the Author):

1. Ventura et al. investigated how multiciliated cell (MCC) precursors integrate into the superficial epithelial layer in the epidermis of Xenopus embryos. MCCs migrate below the epithelium before selecting an integration site and inserting at cell vertices in a process called radial intercalation. While recent studies have begun to shed light on this process, the underlying cellular mechanics was not understood and it was not clear how MCCs select their site of integration into the epithelium.

The authors present a very careful systematic analysis of the dynamic interplay between intercalating MCCs and the surrounding epithelial cells. They show that migrating MCCs extend dynamic filopodia that interact with epithelial cell vertices and thereby “probe” the local mechanical properties of the epithelium. Interestingly, the vertex-specific transmembrane protein LSR is enriched on tips of MCC filopodia as they pull on vertices, suggesting that LSR mediates interactions between filopodia and cell vertices. Consistent with this idea, the authors show that depletion of LSR from MCCs leads to impaired filopodial growth and reduced apical emergence of MCCs. They went on to test the idea that MCC filopodia are able to sense mechanical properties of the epithelium. They employed a theoretical model that simulates out-of-plane pulling forces at vertices by filopodia, yielding a map of local vertex “stiffness”. The model (the mathematical and physical basis of which I am not competent to judge) predicted that the line tension along cell-cell junctions constituting a vertex determines its propensity as a site for radial intercalation, with higher line tensions and resulting higher vertex stiffness favoring intercalation. Furthermore, heterogeneous line tension is predicted to promote the formation of higher-fold vertices where four or more cells meet, and such higher-fold vertices, as opposed to the more prevalent 3-fold vertices, are predicted to favor radial intercalation of MCCs. The authors tested these predictions using a combination of live imaging, cell labeling, and quantitative analyses. Consistent with model, they demonstrate that 4-fold vertices are indeed preferred over 3-fold vertices as sites for MCC insertion. Accumulation of myosin II (as detected using a GFP-labelled myosin II-specific nanobody) along cell-cell junctions is used as a proxy for line tension and vertex stiffness. Here, a more direct way of assessing junctional tension, e.g. using laser ablations, would be necessary to validate the myosin II measurements and to substantiate the authors’ conclusions regarding vertex mechanics, which are key to the study.

We thank the reviewer for carefully reading our work. We have now included laser ablation experiments to directly infer junctional tension, which further supports the use of a myosin II probe as a tension sensor and also predictions of our theoretical model (included in Supplementary Fig. 5). Our data show that both recoil and myosin-II intensity scale with junction length, with shorter junctions showing increased levels of

myosin-II intensity and recoil velocities upon ablation. Moreover, by performing laser ablations, we found no statistical difference in the junctional tension of junctions connected to either 4-fold and 3-fold vertices (Supplementary Fig. 5f), supporting the model prediction that it is predominantly the number of junctions that determines vertex stiffness and consequently the probability of vertex opening.

2. Finally, the authors show that the interaction between integrating MCCs and surrounding epithelial cells drives junctional remodeling and the formation of higher-fold vertices, and that Myosin II activity is required in MCCs for these processes.

Altogether, this is an impressive body of very thorough work, combining live imaging and quantitative analyses with in silico modeling, the predictions of which, in turn, are tested by functional experiments. The results are convincing, carefully quantified and very clearly documented in text and figures. The work provides significant new insights into the mechanics of radial cell intercalation in a developmental context. These new findings are likely to have important implications also for other situations where cells move through tissues, such as transmigration of leukocytes or of metastatic cancer cells through endothelial vessel walls. Hence, the work presents a significant conceptual advance that will be of interest to a broad audience. The manuscript should be accepted for publication, given that the authors address the following points.

We thank the reviewer for a constructive critique. We were delighted that the referee finds our work as “an impressive body of very thorough work” presenting “a significant conceptual advance that will be of interest to a broad audience”. We believe that working through the referee’s comments has further improved our study.

2a. The title of the paper is extremely general and implies that an all-encompassing model for the mechanics of cell integration is presented. However, cell integration events in different tissues and between different cell types are likely to involve at least in part different mechanisms and mechanics (e.g., there may be cell integration events that do not take place at cell vertices). The authors should consider rephrasing the title accordingly.

We have now changed the title to: **"Multiciliated cells use filopodia to probe tissue mechanics during epithelial integration *in vivo*"** to better reflect the focus of our study.

2b. The vertex-based model assumes that higher-order vertices represent direct contacts between 4 or more cells. However, higher-order vertices may in fact represent multiple closely spaced three-fold vertices (tricellular junctions) that cannot be resolved as separate entities by confocal microscopy. The authors should comment on this issue and its possible implications for MCC behavior. If cell vertices provide docking sites for

MCC filopodia, could the presence of multiple closely adjacent docking sites explain why such apparent higher-order vertices provide preferential sites for MCC integration?

Rebuttal Fig.1 - Live Super Resolution Imaging of Rosette formation. A rosette is starting to be formed by the integrating MCC (confocal) which was then imaged using the Super Resolution mode of the Airy Scan module (Airy Scan). While the epithelial vertices can be easily distinguished at timepoints 0 and 19 min, at timepoint 8 min the LSR strings cannot be easily distinguished from the LSR accumulated at the leading edge of the MCC (see inset)

This is an interesting point raised by the reviewer. Work in the *Drosophila* embryo has shown that rosettes of 5, 6, and 7-vertices are, in fact, combinations of several 3-way vertices (Finegan et al., 2019). However, rosette formation in our situation is different, as rosettes are formed by the collapse of different junctions by the intercalating cell, which is positioned right at the center of the rosette. To try to address the reviewer's point of how these structures are formed, we have now performed live Super-Resolution imaging using a Zeiss AiryScan system. We observed that the intercalating MCC forms a complex and highly-transitory structure with the neighboring vertices, with an LSR cluster connecting the intercalating MCC and the goblet cells and where the vertices cannot be easily discernible (Rebuttal Fig. 1). However, it is possible that the optical resolution of our Zeiss AiryScan system could still be insufficient to resolve these structures. This would require applying either STED or STORM for imaging *Xenopus* embryonic epidermis, which unfortunately is not a technically straightforward task. Nevertheless, our preliminary data indicate that the formation of higher-order vertices is triggered by integrating MCCs.

Further experimental and theoretical work would be required to dissect how these transient rosette-like architectures are formed and resolved. There is a growing consensus in the field that rosettes are energetically unstable structures that must

resolve, providing an efficient mechanism to remodel tissues (Yan and Bi 2019). It will be extremely interesting to dissect how higher-order vertices formed by MCCs influence the overall tissue mechanical properties, and how the resolution of these rosette-like structures triggers tissue-scale morphological changes. However, these are complex questions that would require in-depth experimental and theoretical studies, and should form the basis for a separate manuscript.

2c. The use of the anti-myosin II nanobody (SF9-3xGFP) as a measure for junctional tension needs to be validated. What does the SF9 nanobody recognize? Does it interfere with myosin II function? How does the distribution of SF9-3xGFP signals correspond to the distribution of active myosin II (phospho-myosin II)? This should be straightforward to address by immunostainings.

The SF9 intrabody (we had incorrectly described it as a nanobody in the original manuscript, but we have since corrected this) recognizes the non-muscle myosin IIA through a highly conserved epitope, and it has been successfully used across different model organisms with different fluorescent protein tags (*Ciona* - Hashimoto et al., 2015; *Xenopus laevis* - Arnold et al., 2019; *Mus musculus* - Chaigne et al., 2013).

We can confirm that the SF9-3xGFP expression does not interfere with myosin II function, as the SF9-3xGFP sensor recapitulates the phospho-myosin II staining at the actomyosin cortex (Supplementary Fig.5c). Moreover, the SF9-3xGFP myosin sensor is also localized at the base of filopodia (Fig. 5a) and at the cleavage furrow of dividing ectodermal cells (now included as Supplementary Fig.5a), all previously reported locations for active myosin-II (Alieva et al. 2019; Herszterg et al. 2013; Hashimoto et al. 2015). Finally, our data show that the myosin intrabody reflects myosin II activity, as both p-myosin II and the myosin intrabody are found enriched in the shorter (more contractile) epithelial junctions (Supplementary Fig.5e).

2d. Although the idea that homotypic LSR-LSR contacts mediate interactions between MCC filopodia and cell vertices is persuasive, no evidence for a direct LSR-mediated interaction (as claimed by the authors) is provided. Does LSR mediate homophilic cell adhesion? In the absence of such evidence, the authors need to tone-down their statements that "... our data demonstrate that MCCs use filopodia to directly interact with vertices by LSR-LSR mediated contacts" (line 110 and elsewhere).

This is a good point, and we have changed this statement accordingly. We have removed the word homophilic and focused solely on contact formation between filopodia and vertices.

2e. Fig. 1c: the color code is confusing. A more intuitive continuous (unidirectional) color scale should be used.

We have now changed the color-coding to a unidirectional color scale.

2f. Fig. 3e is not showing a probability density function (PDF). $K\delta$ is a continuous (random) variable. As such, the integral of the PDF must equal 1. This is not the case in the presented graph. The authors should use a correct way to estimate the PDF of $K\delta$, for instance by using kernel density estimations. Alternatively, a simple histogram could be shown.

We thank the reviewer for pointing this out. We have now replaced Fig. 3e, representing PDF with the appropriate kernel such that the integral is equal to 1.

2g. Fig. S3f: The graphs are lacking a legend on the Y-axis.

We have included the correct legend in the Y-axis.

References

- Alieva, N. O., A. K. Efremov, S. Hu, D. Oh, Z. Chen, M. Natarajan, H. T. Ong, et al. 2019. "Myosin IIA and Formin Dependent Mechanosensitivity of Filopodia Adhesion." *Nature Communications* 10 (1): 3593.
- Barriga, Elias H., Kristian Franze, Guillaume Charras, and Roberto Mayor. 2018. "Tissue Stiffening Coordinates Morphogenesis by Triggering Collective Cell Migration in Vivo." *Nature* 554 (7693): 523–27.
- Hashimoto, Hidehiko, Francois B. Robin, Kristin M. Sherrard, and Edwin M. Munro. 2015. "Sequential Contraction and Exchange of Apical Junctions Drives Zippering and Neural Tube Closure in a Simple Chordate." *Developmental Cell* 32 (2): 241–55.
- Herszterg, Sophie, Andrea Leibfried, Floris Bosveld, Charlotte Martin, and Yohanns Bellaiche. 2013. "Interplay between the Dividing Cell and Its Neighbors Regulates Adherens Junction Formation during Cytokinesis in Epithelial Tissue." *Developmental Cell* 24 (3): 256–70.
- Higashi, Tomohito, Shinsaku Tokuda, Shin-Ichiro Kitajiri, Sayuri Masuda, Hiroki Nakamura, Yukako Oda, and Mikio Furuse. 2013. "Analysis of the 'Angulin' Proteins LSR, ILDR1 and ILDR2--Tricellulin Recruitment, Epithelial Barrier Function and Implication in Deafness Pathogenesis." *Journal of Cell Science* 126 (Pt 4): 966–77.
- Marchant, C. L., A. N. Malmi-Kakkada, J. A. Espina, and E. H. Barriga. 2021. "Microtubule Deacetylation Reduces Cell Stiffness to Allow the Onset of Collective Cell Migration in Vivo." *bioRxiv*. <https://doi.org/10.1101/2021.08.12.456059>.
- Oda, Yukako, Tetsuhisa Otani, Junichi Ikenouchi, and Mikio Furuse. 2014. "Tricellulin

Regulates Junctional Tension of Epithelial Cells at Tricellular Contacts through Cdc42." *Journal of Cell Science* 127 (Pt 19): 4201–12.

Yamamoto, Kei, Haruko Miura, Motohiko Ishida, Yusuke Mii, Noriyuki Kinoshita, Shinji Takada, Naoto Ueno, Satoshi Sawai, Yohei Kondo, and Kazuhiro Aoki. 2021.

“Optogenetic Relaxation of Actomyosin Contractility Uncovers Mechanistic Roles of Cortical Tension during Cytokinesis.” *Nature Communications* 12 (1): 7145.

Yan, Le, and Dapeng Bi. 2019. “Multicellular Rosettes Drive Fluid-Solid Transition in Epithelial Tissues.” *Physical Review X*. <https://doi.org/10.1103/physrevx.9.011029>.

REVIEWERS' COMMENTS

Reviewer #1 (Remarks to the Author):

The authors have revised the manuscript along the lines suggested by this and all the reviewers of the article. The new data and amendments to figures and text have considerably improved the manuscript and strengthened the conclusions that can be extracted from it. Thus, I fully support this manuscript for publication in Nature communications.

Reviewer #2 (Remarks to the Author):

Dear all, first of all I cannot apologize enough for the delay in sending out my comments.

This is a resubmission. I had a lot of concerns when reading the first version of the manuscript, especially with three aspects:

1/ lack of details on some experiments, especially the lack of rescue and/or proper controls for specificity and efficiency.

2/ overstated conclusions and interpretations (the balance between descriptive and experimental work).

3/ misleading title and abstract that did not frame the work properly.

I have to say that I was not expecting the authors to be able to reverse my decision on this paper but they did. They should be commended for their effort during revision.

Therefore, I have no more comments and now support publication of this paper in its present form.

However, I would like to add a word about experimental design, controls and selection of paper to be reviewed. The first submission of this work was lacking important details about some tools for loss of function and important controls. (the fact that out of 3 reviewers I was the only one to spot problems with the MO experiments is puzzling). Some conclusions were based on unfinished experiments. Yet, it

passed the editorial step. This should not happen. Because, in some cases the will to match initial conclusions when revising a paper may be so strong for some people that they will provide the controls that are expected by the reviewers, no matter what. Maybe cutting corners while doing so. I am not saying that it happened here but I think the first version of this paper, with some of its main conclusions only supported by partially executed experiments, should not have made it to reviewers in the first place. I think that part of the first editorial decision to send the paper out for review may have been based on trendy keywords. It is about mechanical aspects of cell biology influencing cell behavior/movement. Editors should be more cautious and look beyond a fancy title. A trendy topic does not mean that every paper on it is actually well done and interesting for the community.

Reviewer #3 (Remarks to the Author):

In the revised manuscript, the authors added new experimental results and analyses that corroborate their conclusions and significantly strengthen the paper. In particular, the laser ablation experiments (Figure S5) validate the myosin II measurements and substantiate the authors' conclusions regarding vertex mechanics. The revised title nicely encompasses the message of the paper and fits well. Overall, this is a substantial piece of original and convincing work, and the manuscript should be published, given that the authors address the minor remaining comments below.

The authors performed high-resolution imaging to address whether higher-order vertices may actually represent multiple closely spaced three-fold vertices. However, as pointed out by the authors, it is likely that the optical resolution of the AiryScan system used here is still insufficient to fully resolve these structures. Moreover, the LSR-3xGFP marker is not strictly limited to vertices, and some LSR-3xGFP signal is visible along the entire plasma membrane. These considerations suggest that the exact topology of cell vertices (which is likely to be relevant here) may be impossible to resolve by optical microscopy using fluorescent protein markers such as LSR-3xGFP. Hence, the authors should add a note to discuss the possibility that higher-fold-vertices may represent aggregations of multiple closely apposed three-fold vertices, which might generate (mechanical or other?) properties that favor MCC docking or integration.

The authors also validated the SF9-3xGFP intrabody to detect myosin II. They present new data (Figure S5) showing that SF9-3xGFP signals recapitulate the distribution of active myosin II, as detectable by anti-phospho-myosin II antibody staining at the cell cortex. Although the results are consistent with the authors' conclusions, it is still not clear whether the SF9 intrabody recognizes the entire pool of myosin II or only an active subfraction, so caution should be used when referring to "SF9-3xGFP" and "myosin II" interchangeably in the figures and text.

Fig. 3g: replace "vertexes" by "vertices".

We thank the reviewer again for the helpful and enthusiastic comments on this paper. Below, we present an itemized list of changes made in response to these comments.

Reviewer #3 (Remarks to the Author)

In the revised manuscript, the authors added new experimental results and analyses that corroborate their conclusions and significantly strengthen the paper. In particular, the laser ablation experiments (Figure S5) validate the myosin II measurements and substantiate the authors' conclusions regarding vertex mechanics. The revised title nicely encompasses the message of the paper and fits well. Overall, this is a substantial piece of original and convincing work, and the manuscript should be published, given that the authors address the minor remaining comments below.

We appreciate that the reviewer finds our revised manuscript of high quality and suitable for publication.

The authors performed high-resolution imaging to address whether higher-order vertices may actually represent multiple closely spaced three-fold vertices. However, as pointed out by the reviewer, it is likely that the optical resolution of the AiryScan system used here is still insufficient to fully resolve these structures. Moreover, the LSR-3xGFP marker is not strictly limited to vertices, and some LSR-3xGFP signal is visible along the entire plasma membrane. These considerations suggest that the exact topology of cell vertices (which is likely to be relevant here) may be impossible to resolve by optical microscopy using fluorescent protein markers such as LSR-3xGFP. Hence, the authors should add a note to discuss the possibility that higher-fold-vertices may represent aggregations of multiple closely apposed three-fold vertices, which might generate (mechanical or other?) properties that favor MCC docking or integration.

This is a good point, and we have now added a note to the main text along with Supplementary Figure 7 (Airyscan image of rosette-like structure), stating that the optical resolution could be insufficient to resolve high-fold vertices. Thus, these structures could represent aggregations of multiple closely positioned three-fold vertices, as pointed out by the reviewer, and further work is needed to dissect the time-evolution of these structures.

The authors also validated the SF9-3xGFP intrabody to detect myosin II. They present new data (Figure S5) showing that SF9-3xGFP signals recapitulate the distribution of active myosin II, as detectable by anti-phospho-myosin II antibody staining at the cell cortex. Although the results are consistent with the authors' conclusions, it is still not clear

whether the SF9 intrabody recognizes the entire pool of myosin II or only an active subfraction, so caution should be used when referring to “SF9-3xGFP” and “myosin II” interchangeably in the figures and text.

We have now expanded our description of the SF9-3xGFP myosin sensor to: “Specifically, as a readout of junctional tension, we quantified myosin II intensity using a non-muscle myosin II A-specific intrabody (SF9-3xGFP, for simplicity referred as myosin II), which has been previously used as a proxy for active myosin II⁴⁰.”

Fig. 3g: replace “vertexes” by “vertices”.

We corrected this typo.